# PROMPTING TO PROMPT: META-TEMPLATE LEARNING FOR TRANSFERABLE PROMPT OPTIMIZATION

## ABSTRACT

Prompt optimization plays a key role in fully leveraging the capabilities of large language models (LLMs). Despite their respective advantages, offline, online, and hybrid prompt optimization methods all suffer from limited transferability and strong reliance on task-specific data. To systematically resolve these limitations, we propose Prompting to Prompt (PTP), a novel framework for optimizing meta-templates, inspired by the idea of learning to learn. PTP introduces meta-templates as structured intermediate representations that decompose prompts into transferable elements, enabling generalization across diverse task. PTP employs a bi-level optimization process: the inner loop that refines prompt elements for individual samples using gradient feedback and element list guidance, and the outer loop that captures transferable structural patterns by comparing element-level changes before and after inner-loop updates. Instead of learning task-specific features, the outer loop generalizes structural knowledge across tasks, continuously updating meta-template structures and selection strategies. This enables PTP to unify offline and online prompt optimization, supporting task-level and query-level prompt generation without retraining. Extensive experiments on six benchmark datasets show that PTP consistently outperforms state-of-the-art baselines, achieving up to 10.52-point gains on challenging tasks like Arena-hard. These results demonstrate that PTP offers a promising solution for more transferable and efficient prompt optimization.

## 1 INTRODUCTION

Large language models (LLMs) excel in various NLP tasks (Zhao et al., 2023; Bubeck et al., 2023; Li et al., 2025a; Brown et al., 2020), but traditional task-specific model tuning is computationally expensive and impractical for large models. Prompt engineering offers an efficient alternative (Shin et al., 2020; Tang et al., 2025; Trivedi et al., 2025), yet minor prompt variations can drastically affect outputs (Lu et al., 2021; Sclar et al., 2023; Zheng et al., 2023; He et al., 2025; Madaan et al., 2023). Manual prompt design often relies on extensive trial-and-error and domain expertise, making it resource-intensive and challenging to scale. To address these limitations, automated prompt engineering has been proposed to replace manual efforts, enabling LLMs to generate more effective responses (Cui et al., 2025; Yan et al., 2024; Agarwal et al., 2024; Ramnath et al., 2025).

While some prompt optimization methods rely on access to internal state variables of the LLM (Lester et al., 2021; Shin et al., 2020), practitioners typically interact with LLMs through APIs, where such access is unavailable. To address this, three broad categories of methods have been proposed: offline optimization, online optimization, and hybrid optimization. *Offline optimization* (Pryzant et al., 2023; Guo et al., 2023; Opsahl-Ong et al., 2024) refines system prompts for specific tasks using existing datasets. While effective in closed environments, these prompts often overfit and fail to generalize to new tasks. *Online optimization* (Cheng et al., 2023; Zheng et al., 2024) dynamically enhances user prompts during inference by training auxiliary models on paired datasets. Although flexible, this approach requires additional model training and exhibits limited transferability to new domains. *Hybrid optimization* (Zhang et al., 2025) combines the strengths of both offline and online methods. It generates optimized system prompts and complementary instructions during the offline phase to support dynamic user prompt enhancement in real-time. While improving runtime performance to some extent, it still faces challenges in cross-task generalization and reusability.

Figure 1: An example of model questioning based on the PTP method under the offline setting. Given a task description and a query as input, PTP generates the solution by instantiating a meta-template.

Despite recent advancements, a fundamental challenge in prompt optimization remains unresolved: most existing methods lack transferability across tasks and domains. This core limitation manifests in two key aspects:

1) **Insufficient Cross-task Performance:** Existing methods are highly dependent on task-specific data and configurations, leading to significant performance degradation across tasks.

2) **Limited Reusability Efficiency:** Optimized prompts are typically one-off solutions, lacking mechanisms for reuse across offline and online scenarios.

Most existing studies focus primarily on enhancing single-task performance, paying limited attention to the challenges inherent in prompt optimization across multiple tasks. A unified, transferable framework enabling efficient and cost-effective offline and online prompt optimization is therefore urgently needed.

In this paper, we propose Prompting to Prompt (PTP), a novel meta-template optimization framework designed to enhance prompt transferability. The core idea of PTP is to introduce a *meta-template*—a structured intermediate representation that decomposes prompts into transferable elements—and use it to guide a generalizable optimization mechanism. PTP supports both offline training and online inference without requiring task-specific retraining. During training, PTP adopts a bi-level optimization strategy: 1) Inner Loop: Leverages model gradient feedback and a prompt element list to perform semantic-level refinement and restructuring of prompts. Unlike conventional string-level tuning, this allows for structural improvements and expressive prompt variants. 2) Outer Loop: Analyzes structural differences between pre- and post-optimization prompts to identify transferable patterns, during which the element list is also leveraged to optimize prompt composition. This step adjusts the meta-template structure and improves cross-task adaptability by accumulating reusable prompt knowledge. In deployment, PTP can generate task-level or query-specific prompts without additional training. Leveraging the learned meta-templates, PTP can quickly adapt to new tasks and dynamically generate enhanced prompts based on user input.

Our contributions are summarized as follows:

1) We introduce PTP, the first self-improving prompt framework based on meta-template, which mitigates overfitting and significantly improves generalization in prompt optimization.

2) PTP accommodates both offline and online modes of prompt optimization. By leveraging a transferable meta-template mechanism, PTP enables prompt reuse across tasks.

3) We conduct comprehensive evaluations on multiple standard benchmarks and LLMs. Results show that PTP consistently outperforms existing methods, achieving an average 10.52-point improvement on the Arena-hard dataset, demonstrating its effectiveness and robustness.

## 2 RELATED WORKS

### 2.1 PROMPT ENGINEERING

Compared to fine-tuning large language models (LLMs), which requires substantial data and computational resources, prompt engineering provides a more efficient alternative by adjusting only the input without modifying model parameters. Traditional manual prompt engineering, however, requires expert knowledge and often lacks transferability across different models (Li et al., 2025b).

These limitations have driven growing interest in automatic prompt engineering methods (Zhang et al., 2022; Shum et al., 2023; Guo et al., 2023; 2025), which aim to reduce human effort and improve generalization.

Automatic prompt engineering is typically categorized into two main types: soft and hard. Soft prompt engineering (Lester et al., 2021; Qin & Eisner, 2021) optimizes continuous embeddings via gradient descent. However, this approach becomes increasingly impractical at scale due to the high computational cost of gradient computation and the inability to access model internals when using API-restricted LLMs. In contrast, hard prompt engineering (Chung et al., 2024; Pryzant et al., 2023) uses discrete textual instructions to guide model behavior. It is more interpretable, flexible, and compatible with black-box LLMs accessed via APIs. Given these advantages, this work adopts the hard prompt engineering approach.

## 2.2 Hard Prompt Engineering

Recent advances in hard prompt engineering can be classified into offline and online optimization. Offline prompt optimization refines task-specific prompts prior to interacting with the target model. Examples include: Automatic Prompt Engineering (APE) (Zhou et al., 2022), which uses Monte Carlo search to iteratively generate and select high-performing prompt variants based on evaluation scores. Prompt Optimization with Textual Gradients (ProTeGi) (Pryzant et al., 2023), which employs beam search to refine prompts using natural language "gradients". Optimization by PROmpting (OPRO) (Yang et al., 2023), where LLMs act as optimizers, iteratively refining candidate prompts using meta-prompts and evaluation feedback. Online prompt optimization refines prompts dynamically through user or model feedback without retraining. Examples include: Black-Box Prompt Optimization (BPO) (Cheng et al., 2023), which rewrites user inputs using human preference data to align LLM outputs with expectations. Prompt Augmentation System (PAS) (Zheng et al., 2024), which enhances user prompts with additional context or guidance from complementary prompts generated by trained LLMs. Recently, hybrid optimization has emerged as a way to combine the strengths of offline and online approaches. For instance, P3 (Zhang et al., 2025) optimizes system prompts offline and constructs a dataset of optimized user prompts paired with complementary instructions to facilitate online optimization. Despite these advancements, existing prompt optimization methods suffer from poor transferability across tasks and domains. In contrast, our approach introduces meta-template that learns common patterns across tasks, enabling both offline and plug-and-play manner for various tasks.

## 3 Method

### 3.1 Preliminary

#### 3.1.1 Objectives of Prompt Engineering:

Consider datasets $D = \{(D_i, C_i)\}$, where $D_i = \{(x_{ij}, y_{ij})\}_{j=1}^m$ refers to the dataset for task $i(i = 1, 2, ..., n)$. Each $D_i$ consists of pairs of questions $x_{ij}$ and their corresponding answers $y_{ij}$. $C_i$ is the task description associated with $D_i$. Given a black-box task LLM $M_{task}$ (e.g., ChatGPT) that generates responses with input questions $x_{ij}$ from $D_i$, the goal of prompt optimization is to identify the optimal prompt $P^*$ that maximizes the expected average task-specific score over $D_i$, denoted by the scoring function $\varepsilon_i(\cdot)$. This can be formalized as:

$$P^* = \arg\max_P \mathbb{E}_{(x_{ij}, y_{ij}) \sim \mathcal{D}_i} \left[ \varepsilon_i \left( M_{task} \left( x_{ij}; P \right), y_{ij} \right) \right] \tag{1}$$

#### 3.1.2 Meta Prompting & Meta-Template:

Traditional prompt design methods, such as Chain-of-Thought (CoT) Prompting (Wei et al., 2022), often struggle with deep reasoning for complex tasks and lack flexibility and generalization for new tasks.

Meta Prompting (Zhang et al., 2023) demonstrates that prompt structures, independent of task-specific content, can already support a degree of cross-task generalization. By aligning tasks with systematically designed prompt forms, it highlights the importance of structural regularities in facil-

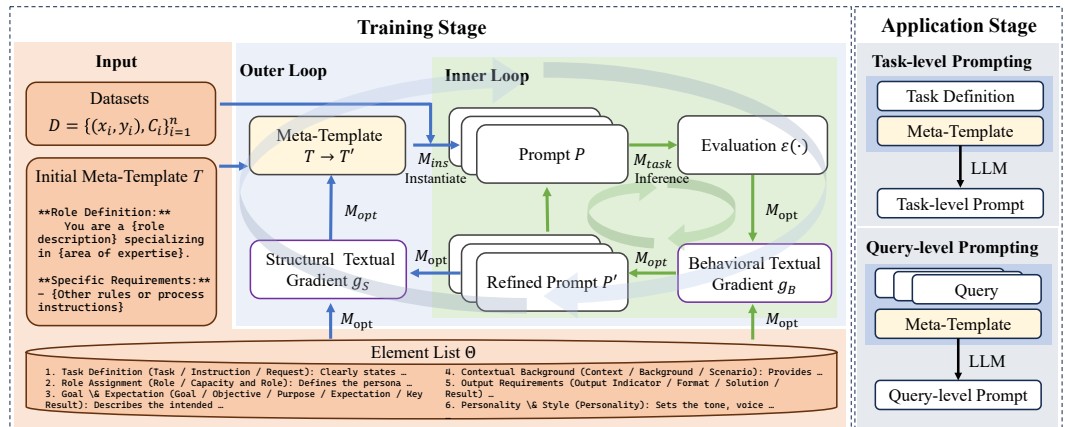

Figure 2: *Overview of the proposed method.* Given a dataset $D$, an initial meta-template $T$, and a static element list $\Theta$—curated from prompt engineering best practices—our framework employs a bi-level optimization to refine $T$ into $T'$. In both the outer loop (updating $T$) and the inner loop (optimizing its instantiated form $P$), elements are iteratively sampled from $\Theta$ to guide structural refinement. At inference time, $T'$ is used to construct prompts in two scenarios: task-level prompting for offline settings, combining task descriptions with $T'$; and query-level prompting for online settings, composing user queries with $T'$. In both cases, the resulting prompts are fed into an LLM for downstream inference.

itating transfer across tasks. Extending this idea, meta-templates build on such inherent generalization by introducing a learning-based mechanism to induce structural patterns from data. Rather than relying on fixed or handcrafted formats, meta-templates capture higher-level abstractions that are optimized through experience, thereby enhancing the robustness and scalability of cross-task transfer. This data-driven extension retains the generalization capacity of Meta Prompting while further improving adaptability in task-level and query-level settings.

### 3.1.3 META-LEARNING & BI-LEVEL OPTIMIZATION:

As a method for optimizing the meta-template, we draw inspiration from meta-learning, often described as "learning how to learn." Designed to improve multi-task generalization by leveraging knowledge across diverse task distributions, meta-learning employs a bi-level optimization process: the inner loop adapts to specific tasks, while the outer loop refines global meta-knowledge to enhance performance across tasks. This approach, exemplified by frameworks like Model-Agnostic Meta-Learning (MAML) (Finn et al., 2017), offers a strong foundation for learning transferable meta-templates that adapt efficiently in multi-task scenarios.

### 3.2 OVERVIEW

Our PTP method introduces a framework to incorporate meta-learning based on meta prompting, as shown in Figure 2. Unlike prior methods (Yang et al., 2023; Pryzant et al., 2023) that focus on individual prompts, PTP learns a transferable meta-template $T$ from task-specific meta-prompts $P$. This structural flexibility allows PTP to identify common patterns, enhancing its transferability across multi-task settings. PTP employs a bi-level optimization process: the inner loop fine-tunes task-specific prompts, while the outer loop refines $T$ by learning a general paradigm across tasks from inner-loop optimization results. The initial meta-template contains only role-defining elements, while the remainder—comprising task-relevant structural patterns and transferable prompt components—is progressively learned and incorporated through optimization. To support meta-template construction, we use an element list $\Theta$—a pool of fundamental prompt components (detailed in Appendix F.1)—as a repository for constructing and refining $T$. Overall, given dataset $D = \{(D_i, C_i)\}$, element list $\Theta$, and initial meta-template $T$ as input, PTP extends Formula 1 into a new optimization objective:

$$T^* = \arg\max_T \mathbb{E}_{(x_{ij}, y_{ij}) \sim \mathcal{D}_i} [\varepsilon_i (M_{task} (x_{ij}; P_i(T, \Theta, C_i)), y_{ij})] \tag{2}$$

where $P_i(T, \Theta, C_i)$ denotes the task-specific prompt instantiated from $T$, $\Theta$ and $C_i$. The optimization is performed by an advanced prompt optimizer $M_{opt}$ (e.g., GPT-4), which generates *textual gradients* (Pryzant et al., 2023; Yuksekgonul et al., 2025)—natural-language feedback analogous to numeric gradients that indicate directions for prompt improvement—to optimize $T$.

Distinct from ProTeGi (Pryzant et al., 2023) that primarily summarizes errors, PTP employs two levels of textual gradients: (1) the inner-loop behavior textual gradient $g_B$, which encodes task-specific feedback for $P_i$ and further suggests structural edits over $\Theta$ (including additions, deletions, and modifications), and (2) the outer-loop structural textual gradient $g_S$, which abstracts cross-task patterns by contrasting pre- and post-optimized prompts to guide meta-template revisions based on $\Theta$. The prompts used during the optimization phase are provided in Appendix F.

The following sections detail the inner and outer loops.

### 3.3 INNER LOOP

For each task $i$, the meta-template $T$ is instantiated into an initial task-specific prompt $P_i^0$ under the task description $C_i$ using an instantiation LLM $M_{ins}$:

$$P_i^0 = M_{ins}(T, C_i; p_{ins}) \tag{3}$$

Here, $p_{ins}$ is the instantiation prompt that $M_{ins}$ requires to generate the task-specific prompt $P_i^0$. During each iteration $t$, the current prompt $P_i^{t-1}$, combined with the input question $x_i$, is used to query the task LLM $M_{task}$, producing predictions $\hat{y}_i$. These predictions are then evaluated against the ground truth $y_i$ with score function $\varepsilon_i(\cdot)$. Errors identified in the evaluation are collected into a set of incorrect examples $\mathcal{B}^{t-1}$:

$$\mathcal{B}^{t-1} \leftarrow Select\left(\{(x, y, \hat{y}) \mid \varepsilon(\hat{y}, y) < \tau\}\right) \tag{4}$$

where $\tau$ is a threshold for selecting incorrect examples. To refine the prompt, the optimizer $M_{opt}$ takes the previous task-specific prompt $P_i^{t-1}$, the error set $\mathcal{B}^{t-1}$, and the element list $\Theta$, guided by the instructive prompt $p_{err}$, and produces behavioral textual gradients $g_B^{t-1}$:

$$g_B^{t-1} = M_{opt}\left(P_i^{t-1}, \mathcal{B}^{t-1}, \Theta; p_{err}\right) \tag{5}$$

These behavioral textual gradients $g_B^{t-1}$ contain insights on prediction errors and element-level editing suggestions for improving the prompt. Guided by $p_{opt}^{inner}$, $M_{opt}$ leverages $g_B^{t-1}$ to refine $P_i^{t-1}$ by performing structural modifications—adding, removing, or revising elements—yielding a series of updated prompts $P_i^t$:

$$P_i^t = M_{opt}\left(P_i^{t-1}, g_B^{t-1}; p_{opt}^{inner}\right) \tag{6}$$

By referencing the element list $\Theta$, this refinement guides the prompt optimization toward meaningful structural and element-level modifications, ensuring systematic and directed improvements.

Following ProTeGi (Pryzant et al., 2023), we apply beam search to select the most promising candidates and use them to initialize the next optimization step. Specifically, at each iteration $t$, multiple candidate prompts $P_i^t$ (from Eq. 6) are evaluated on the validation set, retaining only the top-$b$ candidates (beam width $b$) based on their validation scores $\varepsilon_i(\cdot)$. Details are provided in Algorithm 2 (Appendix B). This inner-loop adaptation is repeated for a fixed number of iterations $K$ (typically a small constant). After $K$ iterations, the refined prompt $P_i^K$ converges to an approximately optimal task-specific prompt $P_i^*$: $P_i^K \approx P_i^*$.

### 3.4 OUTER LOOP

In contrast to the inner loop, which produces behavioral gradients from incorrect examples, the outer loop derives structural gradients by comparing prompts before and after inner-loop refinement. Specifically, it analyzes the changes from the initial prompt $P_i^0$ to the refined prompt $P_i^K$, attributing improvements in task performance to structural modifications. Guided by the element list $\Theta$, this process generates the structural textual gradient:

$$g_S = M_{opt}\left(P_i^0, P_i^K, \Theta; p_{dif}\right) \tag{7}$$

where $p_{dif}$ is an instructive prompt guiding $M_{opt}$ to generate the structural gradient.

**Algorithm 1** PTP Algorithm

**Require:** Dataset $D = \{(x, y), C\}$, Initial Meta-Template $T$, Element List $\Theta$
1: **for** $k \leftarrow 1$ to $K_{Outer} + 1$ **do**
2:    Instantiate $T$ with current tasks: Eq.3;
3:    **for** $t \leftarrow 1$ to $K_{Inner} + 1$ **do**
4:       Generate predictions $\hat{y}$;
5:       Evaluate $\hat{y}$ and generate incorrect example set: Eq. 4;
6:       Yield behavioral textual gradient: Eq. 5;
7:       Refine instantiated prompts: Eq. 6;
8:       Select refined prompts via beam search; {See Alg. 2}
9:    **end for**
10:   Generate structural textual gradient: Eq. 7;
11:   Refine the meta-template: Eq. 8;
12:   $T \leftarrow T'$;
13: **end for**
14: $T^* \leftarrow T$;
15: **return** Optimal Meta-Template $T^*$

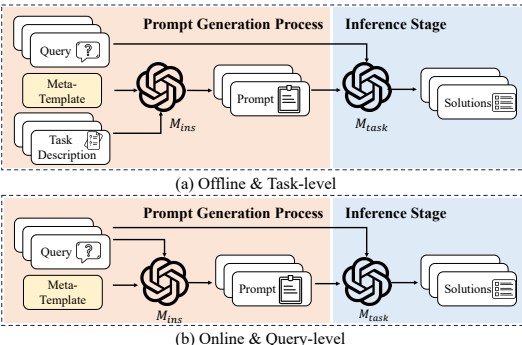

(a) Offline & Task-level

(b) Online & Query-level

Figure 3: Application pipeline of PTP. First, instantiation LLM $M_{ins}$ generates prompts based on the meta-template and task description/user query. These prompts are then input into Task LLM $M_{task}$ to produce solutions.

Similarly, the structural textual gradient $g_S$ guides updates to the meta-template $T$. Structural patterns or elements that consistently improve performance are reinforced, while less effective ones are pruned. The meta-template is updated as:

$$T' = M_{opt}(T, g_S; p_{opt}^{outer}) \tag{8}$$

Through iterative meta-training, $T'$ gradually converges toward an optimal meta-template $T^*$ that encodes generalizable structural priors for prompt construction across tasks: $T' \approx T^*$

The pseudo code of PTP is shown in Algorithm 1.

### 3.5 OFFLINE / ONLINE APPLICATION

In the offline setting (Figure 3(a)), each new task undergoes a single inner-loop adaptation using $T$ to generate a task-specific prompt $P_i$, which the LLM uses to infer solutions based on user queries. In the online setting, the meta-template is adapted in a single step based on the query to produce task-specific prompts for inference. By encoding transferable prompt structures, $T$ enables effective prompts with just one refinement, avoiding the need for extra training (Cheng et al., 2023) or query-specific augmentation models (Zheng et al., 2024).

## 4 EXPERIMENTS

### 4.1 EXPERIMENTAL SETUP

**Datasets.** We evaluate our method on six widely-used benchmarks: GSM8K (Cobbe et al., 2021), MATH (Hendrycks et al., 2021), Big-Bench Hard (BBH) (Suzgun et al., 2022), Arena-Hard (Li et al., 2024), Alpaca-Eval-2.0 (Li et al., 2023), and Alpaca-Eval-2.0-LC (Dubois et al., 2025). For BBH, we select 8 representative tasks spanning diverse domains and reasoning types: Causal Judgment (CJ), Hyperbaton (HB), Logical Deduction Seven Objects (LD), Multistep Arithmetic Two (MA), Navigate (Nav), Reasoning About Colored Objects (CO), Tracking Shuffled Objects Five Objects (Track-5), and Tracking Shuffled Objects Seven Objects (Track-7). These datasets collectively cover arithmetic reasoning, symbolic problem solving, complex task understanding, and general-purpose alignment evaluation. More details about datasets are shown in Appendix C.

**Comparison Methods & Baselines.** To assess the effectiveness of PTP, we compare it against several LLM-based prompting methods employing diverse optimization strategies. These include offline approaches such as APE (Zhou et al., 2022), ProTeGi (Pryzant et al., 2023), and OPRO (Yang et al., 2023); the online method PAS (Zheng et al., 2024); and the hybrid framework P3 (Zhang et al., 2025). We also consider chain-of-thought prompting ("CoT") (Kojima et al., 2022), which

Table 1: Performance comparison of different prompting methods under varying training set sizes for cross-task transferability. Bold highlights the best method per training setting.

| Method | Reasoning | | BBH Tasks | | | | | | | | |
|--------|-----------|------|-----|-----|-----|-----|-----|-----|---------|---------|---------|
| | GSM8K | MATH | CJ | HB | LD | MA | Nav | CO | Track-5 | Track-7 | Average |
| Raw | 93.23 | 75.46 | 62.75 | 86.80 | 59.33 | 96.04 | 94.27 | 83.73 | 68.40 | 68.13 | 77.48 |
| CoT | 94.10 | 77.32 | 63.10 | 86.67 | 57.60 | 98.27 | 92.80 | 84.00 | 71.07 | 66.53 | 77.51 |
| **Training Set: GSM8K** | | | | | | | | | | | |
| ProTeGi | 94.20 | 77.41 | 63.10 | 90.60 | 61.20 | 96.60 | 95.20 | 82.80 | 68.80 | 65.60 | 77.92 |
| APE | 93.75 | 75.03 | 64.71 | 86.00 | 60.40 | 98.00 | 95.20 | 83.20 | 68.40 | 69.60 | 78.06 |
| OPRO | 93.56 | 75.88 | 64.17 | 87.20 | 60.00 | 96.80 | 97.20 | 88.80 | 73.20 | 69.60 | 79.62 |
| PTP | **94.90** | **77.82** | **65.78** | **92.60** | **61.60** | 97.80 | **97.30** | **89.20** | **77.20** | **75.60** | **82.14** |
| **Training Set: BBH Causal Judgement** | | | | | | | | | | | |
| ProTeGi | 93.85 | 76.60 | **67.30** | 89.20 | 54.00 | 97.20 | 91.60 | 84.20 | 67.60 | 70.00 | 77.69 |
| APE | 93.56 | 73.41 | 64.71 | 87.60 | **63.20** | 96.20 | 94.40 | 81.60 | 70.00 | 70.80 | 78.56 |
| OPRO | 93.93 | 72.37 | 64.17 | 91.20 | 62.40 | 92.40 | **95.60** | 82.40 | 71.20 | 69.40 | 78.60 |
| PTP | **94.76** | **77.43** | 66.67 | **92.40** | **63.20** | **98.60** | 95.00 | **84.60** | **75.60** | **76.00** | **81.33** |

uses the instruction "*Let's think step by step.*". Additionally, original LLM outputs without prompt optimization are reported as baselines.

**Implementation Details.** In the offline setting, GPT-4o-mini serves as both the the instantiation model ($M_{ins}$) and task model ($M_{task}$) for all methods, while prompt optimization is performed by GPT-4o ($M_{opt}$). The initial meta-template (Appendix G.2) includes only role-defining elements, with structural components learned progressively from element list $\Theta$ (Appendix F.1) through optimization. To ensure consistent initialization, all methods use a standardized prompt: "*Solve the problem step by step and provide the final answer.*", which PTP uses to instantiate the initial meta-template. PTP sets both inner-loop and outer-loop iterations to 4, as supported by the ablation in Figure 5. In the online setting, no training is required. The optimized meta-template $T^*$ is instantiated on-the-fly using $M_{ins}$ with user queries, and inference is conducted by $M_{task}$ using the resulting prompt. To evaluate the generalization of $T^*$ across different LLMs, we test on a diverse set: Qwen2-7B-Instruct (Bai et al., 2023), GPT-4-turbo-2024-04-09, GPT-4-1106-preview, GPT-4o-mini, GPT-3.5-turbo-1106 (Welsby & Cheung, 2023), and DeepSeek-V3-0324 (Liu et al., 2024). Each model serves as both $M_{ins}$ and $M_{task}$ during inference. Accuracy is reported and averaged over three runs. Additional implementation details are provided in Appendix D.

## 4.2 MAIN RESULTS

We evaluate whether Prompting to Prompt (PTP) can learn generalizable meta-templates that improve performance on unseen tasks. Specifically, we test if meta-templates trained on a limited task corpus (e.g., GSM8K or BBH-CJ) exhibit strong cross-task transferability. We further examine whether these templates can be directly deployed in online settings without requiring any re-training, while still maintaining significant performance gains over existing prompting methods. These experiments aim to validate the scalability and practical applicability of PTP across diverse tasks.

### 4.2.1 OFFLINE.

To evaluate the zero-shot transferability of PTP meta-templates, we conduct offline experiments training on a single dataset, either GSM8K or BBH Causal Judgment (CJ), and testing across three benchmarks: GSM8K, MATH, and eight diverse BBH tasks. As shown in Table 1, when trained on GSM8K, PTP achieves 94.9% on GSM8K, 77.82% on MATH, and an average of 82.14% on BBH tasks, surpassing the best baseline (OPRO) by 2.52 points on BBH and 1.94 points on MATH. Similarly, when trained on BBH-CJ, PTP obtains 94.76%, 77.43%, and 81.33%, again outperforming all baselines in terms of average BBH accuracy, with a +3.64 point gain over ProTeGi and +2.73 points over OPRO. These results demonstrate that PTP can extract reusable structural patterns from a single task and generalize effectively across domains. PTP consistently outperforms strong baselines—Zero-shot CoT, ProTeGi, APE, and OPRO—highlighting its advantage in capturing transferable prompt structures.

Table 2: Comprehensive performance comparison of online prompting methods using transferred meta-templates trained on GSM8K across multiple models and benchmark datasets (Arena-hard, Alpaca-Eval-2.0, Alpaca-Eval-2.0-LC), demonstrating consistent and significant improvements without additional training.

| Dataset | Model | Raw | PAS | P3 | PTP | Δ |
|---|---|---|---|---|---|---|
| Arena-hard | GPT-4-turbo-2024-04-09 | 76.60 | 76.90 | 78.00 | **84.60** | +6.60 |
| | GPT-4-1106-preview | 74.80 | 78.80 | 77.21 | **85.20** | +6.40 |
| | GPT-3.5-turbo-1106 | 18.90 | 22.10 | 25.56 | **47.50** | +21.94 |
| | Qwen2-72b-Instruct | 48.10 | 52.20 | 52.82 | **69.60** | +16.78 |
| | DeepSeek-V3-0324 | 92.50 | 93.00 | 92.70 | **94.00** | +1.00 |
| | GPT-4o-mini | 73.80 | 72.50 | 73.30 | **81.80** | +8.00 |
| | **Average** | 64.12 | 65.92 | 66.60 | **77.12** | +10.52 |
| Alpaca-Eval-2.0 | GPT-4-turbo-2024-04-09 | 46.12 | 65.31 | 70.50 | **70.62** | +0.12 |
| | GPT-4-1106-preview | 50.00 | 65.92 | 69.94 | **70.19** | +0.25 |
| | GPT-3.5-turbo-1106 | 9.20 | 15.82 | 34.53 | **50.94** | +16.41 |
| | Qwen2-72b-Instruct | 31.70 | 45.53 | 61.37 | **62.38** | +1.01 |
| | DeepSeek-V3-0324 | 74.84 | 79.06 | 80.25 | **81.25** | +1.00 |
| | GPT-4o-mini | 41.31 | 56.75 | 59.75 | **69.38** | +9.63 |
| | **Average** | 42.20 | 54.73 | 62.72 | **67.46** | +4.74 |
| Alpaca-Eval-2.0-LC | GPT-4-turbo-2024-04-09 | 55.02 | 56.54 | 58.31 | **62.60** | +4.29 |
| | GPT-4-1106-preview | 50.00 | 53.63 | 56.95 | **59.91** | +2.96 |
| | GPT-3.5-turbo-1106 | 19.30 | 23.31 | 35.35 | **49.65** | +14.30 |
| | Qwen2-72b-Instruct | 39.24 | 44.31 | 55.72 | **56.58** | +0.86 |
| | DeepSeek-V3-0324 | 78.25 | 78.29 | 79.04 | **80.64** | +1.60 |
| | GPT-4o-mini | 49.82 | 51.51 | 51.74 | **58.01** | +6.27 |
| | **Average** | 48.61 | 51.27 | 56.19 | **61.23** | +5.04 |

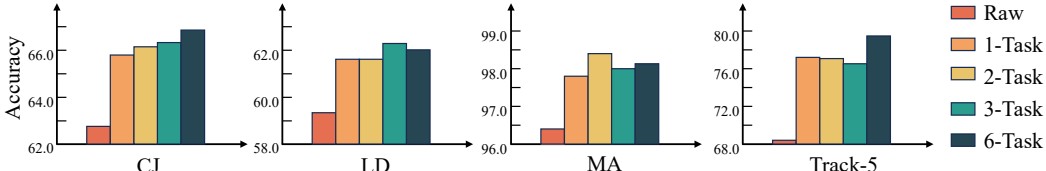

Figure 4: Effect of multi-task training scale on PTP's transfer accuracy across BBH evaluation tasks.

### 4.2.2 ONLINE.

To evaluate the plug-and-play capability of PTP across different large language models (LLMs), we apply a GSM8K-trained meta-template to three open-ended benchmarks—Arena-hard, Alpaca-Eval 2.0, and Alpaca-Eval 2.0 (LC). For each LLM, we keep the task model and the instantiation model identical, ensuring a fair comparison without any re-training or adaptation. As shown in Table 2, PTP significantly outperforms all online baselines—including Raw, PAS, and P3—achieving average improvements of 4.74 to 10.52 points over P3, the strongest baseline.

### 4.3 DETAIL ANALYSIS

We conduct a detailed analysis of the key factors that affect the transferability of PTP. In particular, we examine the role of task descriptions in prompt instantiation, the effect of multi-task training, and the influence of the prompt instantiation model.

**Effect of Multi-task Training Scale.** Figure 4 presents the impact of the number of training tasks on PTP's transfer performance across BBH evaluation tasks. As the number of training tasks increases from 1 to 6, the overall generalization of meta-templates is maintained or improved. Specifically, accuracy on Track-5 improves from 77.20% with a single-task PTP to 79.47% with six tasks, indicating that exposure to a broader range of tasks allows the model to capture more transferable patterns. These results suggest that exposing PTP to a broader range of tasks enables it to capture more transferable patterns and adapt more effectively to unseen tasks. Although performance on a

Table 3: Effect of instantiation model selection on PTP prompt generation performance across datasets, with a fixed GPT-4o-mini task model. Raw scores are reported once per dataset, while PTP and $\Delta$ are shown for each instantiation model.

| Dataset | Raw | GPT-3.5-turbo-1106 | | GPT-4o-mini | | DeepSeek-V3 | |
|---|---|---|---|---|---|---|---|
| | | PTP | $\Delta$ | PTP | $\Delta$ | PTP | $\Delta$ |
| Arena-hard | 73.80 | 79.76 | **+5.96** | 81.81 | **+8.01** | 81.33 | **+7.53** |
| Alpaca-Eval 2.0 | 41.31 | 68.62 | **+27.31** | 69.38 | **+28.07** | 64.56 | **+23.25** |
| Alpaca-Eval 2.0 (LC) | 49.82 | 55.89 | **+6.07** | 58.01 | **+8.19** | 57.66 | **+7.84** |

few evaluation tasks may fluctuate as more training tasks are introduced, the overall trend remains consistent—greater task diversity leads to more generalized prompt behavior.

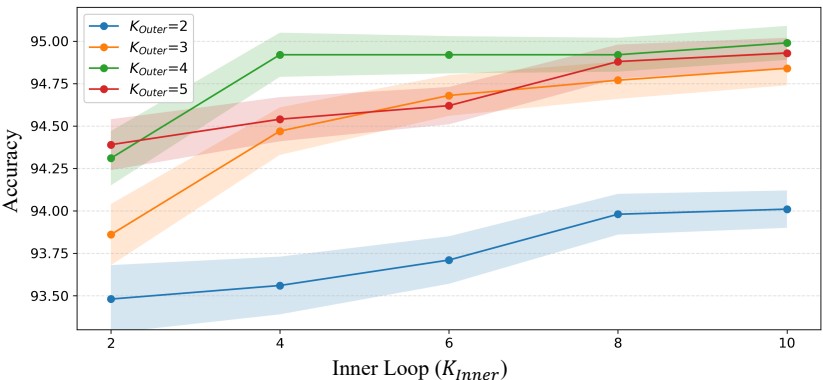

Figure 5: Ablation study on the number of inner and outer loops in PTP optimization.

**Consistency of the Instantiation Model.** To evaluate the consistency of downstream task performance to the instantiation model selection, we fix the task model (GPT-4o-mini) and instantiate prompts using three models: GPT-3.5-turbo-1106, GPT-4o-mini, and DeepSeek-V3. As reported in Table 3, all three instantiation models achieve comparable improvements over the Raw baseline across datasets. For instance, on Arena-hard, PTP instantiated with GPT-3.5-turbo-1106 achieves 79.76% (+5.96), closely matching GPT-4o-mini (81.81%, +8.01) and DeepSeek-V3 (81.33%, +7.53). Similar patterns are observed on Alpaca-Eval 2.0 and its long-context variant, with performance differences limited to a few percentage points. These results indicate that PTP meta-templates are largely insensitive to the choice of instantiation model, maintaining strong transferability without requiring models with greater capacity. This highlights the practical robustness of PTP, enabling effective prompt generation across diverse instantiation scenarios.

**Ablation on Loop Iterations.** To assess the impact of iteration numbers in PTP's bi-level optimization, we systematically varied the outer-loop ($K_{Outer}$) and inner-loop ($K_{Inner}$) iterations. Results are shown in Figure 5. Performance generally improves with larger $K_{Outer}$ or $K_{Inner}$, but the gains plateau beyond 4. Increasing $K_{Inner}$ from 2 to 4 leads to substantial improvements across all $K_{Outer}$ values, but further increases yield diminishing returns. Similarly, raising $K_{Outer}$ from 2 to 4 steadily improves performance for each $K_{Inner}$, though performance slightly degrades beyond 4. Based on this trend, we set $K_{Outer} = 4$ and $K_{Inner} = 4$ as the default to balance performance and efficiency, enabling sufficient prompt refinement and cross-task adaptation without excessive computation.

## 5 CONCLUSION

In this paper, we propose Prompting to Prompt (PTP), a novel framework that optimizes meta-templates to enhance the transferability of prompt optimization across tasks. By introducing meta-templates as structured intermediate representations, PTP enables the decomposition of prompts into

transferable elements and supports a bi-level optimization strategy that jointly refines prompt content and structure. Unlike existing methods that are often task-specific or costly to adapt, PTP offers a unified solution for both offline and online scenarios, requiring no additional training at deployment time. Extensive experiments across multiple benchmarks demonstrate that PTP consistently outperforms existing approaches, particularly in challenging transfer settings. The results demonstrate that PTP is both effective and practical, offering a scalable, low-cost, and reusable framework for prompt optimization.

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

APPENDIX

# A   THE USE OF LARGE LANGUAGE MODEL

Large language model (LLM) was used for language refinement and literature search during manuscript preparation.

# B   BEAM SEARCH OVER PROMPTS

The beam search procedure used in the inner loop (Section 3.3) is formally described in Algorithm 2. This search strategy guides the exploration of prompt candidates generated via behavioral gradient refinement (Eq. 6).

---

**Algorithm 2** Beam Search for Prompt Selection

---

**Require:** Candidates $C$ (all $P_i^t$ in Eq. 6), Validation set $D_i^{val}$, Beam width $b$, Metric function $\varepsilon(\cdot)$
1: **for** each candidate $c \in C$ **do**
    {Evaluate candidates}
2:     Compute validation score: $s(c) \leftarrow \mathbb{E}_{(x,y) \sim D^{val}}[\varepsilon(M_{task}(x; c), y)]$
3: **end for**
4: Rank candidates by score: $C_{ranked} \leftarrow \text{Sort}(C, \text{by} = s(\cdot), \text{descending})$
5: Select top-$b$ candidates: $B \leftarrow C_{ranked}[1 : b]$
6: **return** Beam set $B$ containing top-$b$ prompts

---

# C   DATASET AND TASK DETAILS

**GSM8K.** GSM8K is a high-quality benchmark containing approximately 8,500 diverse grade-school level math word problems. It is specifically designed to evaluate the arithmetic and reasoning capabilities of language models through detailed, step-by-step problem solving. In our experiments, we utilize 200 instances from the training set and all 1,319 instances from the test set to evaluate model performance on multi-step quantitative reasoning tasks.

**MATH.** MATH is a carefully curated subset of 500 challenging problems from the larger MATH dataset, covering topics such as algebra, geometry, and calculus. It is designed to rigorously evaluate advanced language models' mathematical reasoning and problem-solving abilities through precise, step-by-step solutions. In our experiments, we utilize 200 instances from the training set and 300 instances from the test set to assess model performance on this high-difficulty benchmark.

**Big-Bench Hard (BBH).** BBH is a challenging subset of the BIG-Bench benchmark, containing tasks selected for their difficulty and ability to evaluate large language models. It spans diverse domains, including logical reasoning, mathematical problem solving, and abstract language understanding, serving as a rigorous testbed for assessing advanced reasoning and generalization capabilities.

We select 8 representative tasks from BBH, covering diverse domains and reasoning types. For each task, we follow a 2:3 split for training and testing. Specifically:

1. **Causal Judgment (CJ)**: Read a short story involving moral, intentional, or counterfactual elements, and determine how an average person would answer a causal question about it. We use 74 instances for training and 113 instances for testing.

2. **Hyperbaton (HB)**: Given two English sentences, identify which one has the correct adjective order. We use 100 instances for training and 150 instances for testing.

3. **Logical Deduction (LD)**: Deduce the order of objects based on clues about their spatial relationships. We use 100 instances for training and 150 instances for testing.

4. **Multi-Step Arithmetic (MA)**: Solve multi-step math problems involving basic arithmetic operations like addition, subtraction, multiplication, and division. We use 100 instances for training and 150 instances for testing.

5. **Navigate (Nav)**: Determine whether an agent returns to its starting point after following a sequence of movement instructions. We use 100 instances for training and 150 instances for testing.

6. **Reasoning About Colored Objects (CO)**: Given a context, answer a simple question about the color of an object on a surface. We use 100 instances for training and 150 instances for testing.

7. **Tracking Shuffled Objects – Five Objects (Track-5)**: Track the final positions of 5 objects after a series of pairwise swaps starting from known initial positions. We use 100 instances for training and 150 instances for testing.

8. **Tracking Shuffled Objects – Seven Objects (Track-7)**: Similar to Track-5, but with 7 objects, increasing the complexity of the tracking task. We use 100 instances for training and 150 instances for testing.

**Arena-hard.**   Arena-hard is a high-difficulty benchmark designed to evaluate the robustness of large language models in complex, ambiguous, and adversarial scenarios. It consists of carefully selected challenging tasks that go beyond conventional evaluation sets, requiring advanced reasoning, problem-solving, and fine-grained language understanding capabilities. This benchmark emphasizes the assessment of models' generalization and alignment under extreme conditions, serving as a key reference for exploring the upper bounds of model performance.

**Alpaca-Eval-2.0** & **Alpaca-Eval-2.0-LC.**   **Alpaca-Eval-2.0** is an enhanced evaluation framework for instruction-following language models, covering diverse tasks and domains to comprehensively assess models' adherence to instructions and alignment quality through open-ended user queries. Its length-controlled variant, **Alpaca-Eval-2.0-LC**, introduces a regression-based debiasing method to mitigate the influence of response length on evaluation scores, thereby improving robustness and interpretability. Leveraging GPT-4 as an automatic annotator, this framework demonstrates both high efficiency and consistency, while maintaining strong correlation with human judgments, making it a reliable tool for automated benchmarking.

**Data Splitting.** Each task's training set is equally divided into training and validation subsets. The inner loop generates candidate prompts via textual gradient updates on the training subset, then evaluates these candidates on the validation subset. Beam search (Algorithm 2) selects the top-b candidates based on validation performance. All reported results are evaluated on separate held-out test sets, completely disjoint from both training and validation data.

# D   SUPPLEMENTARY EXPERIMENTAL DETAILS (MAIN TEXT)

In all experiments, the temperature of both the task model ($M_{task}$) and the instantiation model ($M_{ins}$) is fixed at 0 to promote deterministic behavior. In contrast, the prompt optimizer ($M_{opt}$) employs a higher temperature of 0.7 to encourage diversity in candidate generation during meta-template refinement. At each step, the optimizer generates 4 candidate prompts.

PTP is trained with $K_{inner} = 4$ inner-loop iterations and $K_{outer} = 4$ outer-loop iterations, refining the meta-template through iterative optimization. For fair comparison, the baseline methods APE, ProTeGi, and OPRO are each trained for 16 iterations, ensuring that all approaches consume computational resources at a comparable scale.

## D.1   EFFECT OF MULTI-TASK TRAINING SCALE.

Figure 4 presents the impact of the number of training tasks on PTP's transfer performance across BBH evaluation tasks. To systematically examine this effect, PTP is trained on progressively larger sets of datasets:

- **1-task setting:** GSM8K only.
- **2-task setting:** GSM8K and Hyperbaton.
- **3-task setting:** GSM8K, Hyperbaton, and Reasoning About Colored Objects.
- **6-task setting:** GSM8K, Causal Judgment, Logical Deduction Seven Objects, Multistep Arithmetic Two, Navigate, and Tracking Shuffled Objects Five Objects.

All training conditions are kept identical to those in the offline phase, ensuring comparability across different scales. This design isolates the effect of task diversity, allowing us to directly assess how broader task exposure influences the meta-template's ability to generalize to unseen evaluation tasks.

# E    SUPPLEMENTARY EXPERIMENTS

## E.1    IMPACT OF TRAINING DATA SELECTION ON PTP META-TEMPLATE TRANSFERABILITY

Table 4: Impact of training set choice on PTP transfer performance compared to initial prompt and initial meta-template baselines.

| Method | Reasoning | | BBH Tasks | | | | | | | | |
|---|---|---|---|---|---|---|---|---|---|---|---|
| | GSM8K | MATH | CJ | HB | LD | MA | Nav | CO | Track-5 | Track-7 | Average |
| Initial Prompt | 93.80 | 76.80 | 65.24 | 90.80 | 61.20 | 97.20 | 96.80 | 85.60 | 64.00 | 62.40 | 79.38 |
| Initial Meta-template | 94.00 | 76.80 | 65.51 | 91.80 | 60.00 | 96.60 | 95.40 | 83.40 | 69.20 | 63.80 | 79.65 |
| **Training Set: GSM8K** | | | | | | | | | | | |
| PTP | 94.90 | 77.80 | 65.78 | 92.60 | 61.60 | 97.80 | 97.30 | 89.20 | 77.20 | 75.60 | 82.14 |
| **Training Set: BBH Causal Judgement** | | | | | | | | | | | |
| PTP | 94.76 | 77.40 | 66.67 | 92.40 | 63.20 | 98.60 | 95.00 | 84.20 | 75.60 | 76.00 | 81.33 |

We analyze the impact of different training datasets on the performance and transferability of PTP-optimized meta-templates. Table 4 reports results for meta-templates trained on GSM8K or the BBH Causal Judgment (CJ) dataset, evaluated across GSM8K, MATH, and eight selected BBH subtasks.

Starting with the baselines, the **Initial Prompt** achieves an average accuracy of 79.38% on BBH subtasks. The **Initial Meta-template**, trained solely on GSM8K, slightly improves performance to 79.65%, suggesting that while single-task templates provide limited structural benefit, their generalization remains constrained. This comparison highlights that structural initialization alone is insufficient for robust cross-task transfer.

Optimizing the meta-template with PTP on GSM8K yields substantial gains. The resulting meta-template reaches 82.14% average accuracy on BBH subtasks, with notable improvements on GSM8K and MATH benchmarks. This demonstrates that iterative PTP training effectively refines prompt structures to capture reasoning patterns across quantitative and logical tasks.

Training PTP on the BBH Causal Judgment dataset also leads to strong performance. The meta-template achieves 81.33% average accuracy on BBH subtasks—slightly below the GSM8K-trained version but well above both baselines. It achieves its highest score on the Causal Judgment task itself (66.67%) and performs strongly on related subtasks such as Logical Deduction and Multistep Arithmetic, indicating that PTP can leverage task-specific features while preserving generalization.

Overall, across both training sets, PTP consistently outperforms baseline approaches. The comparison between Initial Prompt and Initial Meta-template underscores that structural guidance alone offers limited benefit, whereas PTP training substantially enhances transferability and reasoning capability, validating its effectiveness for zero-shot adaptation across heterogeneous reasoning tasks.

## E.2    IMPACT OF TASK-AWARE PROMPT INSTANTIATION ON CROSS-TASK TRANSFERABILITY

Table 5: Effect of task description and example augmentation during prompt instantiation on BBH transfer.

| Method | With Task Description | With Example | CJ | LD | MA | Track-5 |
|---|---|---|---|---|---|---|
| Raw | | | 62.75 | 59.33 | 96.40 | 68.40 |
| ProTeGi | | | **67.30** | 64.33 | 97.20 | 74.08 |
| PTP | | | 65.78 | 61.60 | **97.80** | 77.20 |
| PTP | ✓ | | 67.18 | 68.40 | 97.33 | 81.30 |
| PTP | ✓ | ✓ | 66.44 | **69.73** | 97.76 | **82.73** |

We investigate whether incorporating task descriptions and examples during prompt instantiation can enhance cross-task transferability. In standard instantiation, prompts are generated solely based

on a meta-template trained on GSM8K, without task-specific guidance. To improve alignment with target tasks, we augment the instantiation process with natural language task descriptions corresponding to each BBH task.

For instance, for the causal reasoning task, the task description reads: *"Given a short story involving moral, intentional, or counterfactual analysis, determine how a typical person would answer a causal question about the story."* This description provides high-level guidance that aligns the prompt with the task's reasoning requirements.

Beyond descriptions, we also incorporate task-specific examples during instantiation to further guide the model. An example for the causal judgment task is as follows:

> **Input:** How would a typical person answer each of the following questions about causation?
> An intern is taking care of a patient in a hospital. The intern notices that the patient is having some kidney problems. ... Despite this policy, the doctor decides to sign off. Since both signatures were received, the patient is administered the drug. As it happens, the patient immediately recovers, and the drug has no adverse effects.
> **Question:** Did the pharmacist's decision cause the patient's recovery?
> **Options:** Yes / No
> **Answer:** No

As shown in Table 5, task-aware instantiation with descriptions consistently improves performance across BBH tasks. Incorporating both task descriptions and examples further enhances the instantiated prompts, yielding additional gains. For example, on the logical deduction (LD) task, accuracy improves by 8.1% compared to instantiation without task guidance. Notably, with this simple augmentation, PTP matches or surpasses ProTeGi, which is trained directly on the target tasks.

## F    ELEMENT LIST AND PROMPTS USED DURING OPTIMIZATION

This section presents the element list utilized in the PTP framework, along with the system prompts required for each LLM, as illustrated in Box F.1 to F.6. Note that these prompts remain fixed during the optimization process.

---

**Box F.1: Element List $\Theta$**

1. Task Definition (Task / Instruction / Request)
Clearly states the specific task or request the model is expected to perform.

2. Role Assignment (Role / Capacity and Role)
Defines the persona or expertise the model should assume to guide tone, knowledge, and behavior.

3. Goal & Expectation (Goal / Objective / Purpose / Expectation / Key Result)
Describes the intended outcome or objective, along with any measurable expectations for the output.

4. Contextual Background (Context / Background / Scenario)
Provides necessary background information or situational context to help the model understand the task accurately.

5. Output Requirements (Output Indicator / Format / Solution / Result)
Specifies the desired form, structure, tone, or type of output expected from the model.

6. Personality & Style (Personality)
Sets the tone, voice, or emotional character of the response—e.g., friendly, concise, formal.

7. Examples & Demonstrations (Examples)
Offers sample inputs and/or outputs to illustrate desired response patterns or behaviors.

8. Action Steps (Action / Steps)
Directs the model through sequential actions or procedures necessary to complete the task.

9. Clarity & Focus (Clarity / Avoid Ambiguity)
Ensures the task is framed clearly and unambiguously, removing any extraneous or misleading information.

10. Relevant Information (Relevant Info)
Supplies supporting details, such as keywords, facts, audience characteristics, tone, or formatting preferences.

11. Insight & Depth (Insight)
Adds deeper context, reasoning, or domain-specific considerations to enrich the model's understanding and response.

12. Structure & Format (Structure / F)
Specifies the organizational format of the response—e.g., bullet points, tables, sections, Markdown.

---

**Box F.2: Instantiation Prompt $p_{ins}$ Required by the Instantiation LLM $M_{ins}$ (Eq. 3)**

You are an expert prompt engineer. Your task is to generate a concrete, executable prompt instance (a grounded prompt) based on the provided "meta-template" and "task description."

Objectives
1. The grounded prompt must be highly executable, directly usable by large language models to significantly improve output quality;
2. Retain the structure and format of the meta-template;
3. Clearly fill in or revise vague or abstract parts of the meta-template to closely reflect the real intent expressed in the user instruction;
4. Avoid vague or generic expressions; the language must be clear, specific, and cover core constraints or expectations;
5. Wrap the generated grounded prompt within ¡START¿ and ¡END¿ tags for easy extraction.

Meta-template
{meta_prompt}

Task Description
{instruction}

Please generate the grounded prompt accordingly:
<START>
(your grounded prompt here)
<END>

**Box F.3: Instructive Prompt $p_{err}$ for Guiding the Optimizer $M_{opt}$ to Generate Behavioral Text Gradients in the Inner Loop (Eq. 5)**

I'm trying to refine a prompt for a large language model.

My current prompt is:
{prompt}

But this prompt gets the following examples wrong:
{error_string}

List of meta-prompt structural elements (add with caution to avoid overly long prompts):
{meta_prompt_elements}

Focus on the difference between ground truth and prediction, and give {num_feedbacks} reasons why the prompt could have gotten these examples wrong.

Based on the element list above, analyze which elements should be added, removed, or modified to improve the effectiveness of the prompt.

Wrap the combined reason-and-edit suggestion within <START> and <END>.

**Box F.4: Instructive Prompt $p_{dif}$ for Guiding the Optimizer $M_{opt}$ to Generate structural Text Gradients in the Outer Loop (Eq. 7)**

You are a prompt engineering expert, specializing in reverse-optimizing prompt generation rules (*meta-prompts*) through fine-grained analysis of concrete prompt templates.

Please conduct an in-depth analysis of the following content:

1. Meta Prompt (meta_prompt)
{meta_prompt}

2. Initial Concrete Prompt & Score
Initial Prompt: {prompt_concrete}
Initial Score: {prompt_concrete_score}

3. Optimized Concrete Prompt & Score
Optimized Prompt: {opt_concrete_prompt}
Optimized Score: {opt_concrete_prompt_score}

4. List of meta-prompt structural elements
{meta_prompt_elements}

Your tasks:

- Identify the key differences between the initial and optimized prompts.
- Based on these differences and the change in scores, reflect deeply and answer:
    1. Which parts of the meta-prompt failed to effectively guide the generation of high-quality concrete prompts? Why?
    2. What successful expression strategies or structural improvements in the optimized prompt could be integrated back into the meta-prompt to enhance its performance?

Your optimization advice should be concise yet insightful, and must include:

- Key points of difference analysis
- Optimization suggestions (in the form: "*You should... because...*")
- Scope of applicability and potential risks
- Wrap your suggestions within <START> and <END> tags for easy extraction

Your goal is to produce thoughtful, generalizable optimization advice for improving meta-prompts.

Please generate your optimization advice as follows:
<TART>
(Insert your optimization advice here)
<END>

**Box F.5: Optimization Prompt $p_{opt}^{inner}$ for Guiding the Optimizer $M_{opt}$ to Generate Better Prompts in the Inner Loop (Eq. 6)**

I'm trying to refine a prompt for large language model.

My current prompt is:
{prompt}

But it gets the following examples wrong:
{error_str}

Based on these examples the problem with this prompt is that {feedback_str} Based on the above information, I wrote a different improved prompt. The improved prompt should be wrapped with <START> and <END>.

The new prompt are:

> **Box F.6: Optimization Prompt $p_{opt}^{outer}$ for Guiding the Optimizer $M_{opt}$ to Generate Better Meta-Template in the Outer Loop (Eq. 8)**
>
> You are a meta-prompt design expert, skilled at integrating evaluation data and structural components to generate more effective meta-prompts. Your task is to optimize the current meta-prompt based on the provided information.
>
> 1. Current meta-prompt
> {current_meta_prompt}
>
> 2. Optimization suggestions
> {optimization_advice}
>
> Please conduct a multi-angle analysis and focus on the following questions:
> - Which optimization suggestions can be transformed into concrete, actionable template content?
> - Which current template elements should be strengthened, weakened, or removed?
> - Should new structural elements be introduced to improve performance? Please be cautious: only propose additions that are truly necessary, and avoid making the prompt unnecessarily long.
>
> Your output should include:
> - A brief explanation of the overall meta-prompt optimization logic, enclosed within <optimization_logic> tags.
> - The fully optimized meta-prompt, enclosed within <optimized_meta_prompt> tags. Do not include any optional elements, every instruction must be essential, explicit, and necessary.
>
> <optimization_logic>
> (Insert a brief explanation of the overall meta-prompt optimization logic here)
> </optimization_logic>
>
> <optimized_meta_prompt>
> (Insert the optimized meta-prompt content here)
> </optimized_meta_prompt>

## G  INITIAL PROMPT AND META-TEMPLATE

This section shows the initial meta-template to be optimized.

> **Box G.1: Initial Prompt**
>
> Solve the problem step by step and provide the final answer.

> **Box G.2: Initial Meta-Template $T$**
>
> **Role Definition:**
> You are a {role description} specializing in {area of expertise}.
>
> **Specific Requirements:**
> - {Other rules or process instructions}

## H  DISPLAY AND COMPARISON OF OPTIMIZED PROMPTS UNDER DIFFERENT EXPERIMENTAL SETUPS

### H.1  OPTIMIZED PROMPTS OF DIFFERENT OFFLINE METHODS

This section presents examples of prompts optimized by our PTP approach and various offline methods on GSM8K dataset, corresponding to Table 1 in the main text.

## Box H.1.1: Offline: Refined Prompt under PTP(Ours) on GSM8K Dataset

**Role Definition:**
You are a highly specialized {role description}, with expertise in {specific area of expertise}. Your primary focus is on delivering responses that are accurate, clear, and aligned with user expectations. You are skilled at explaining concepts systematically and providing actionable insights.

**Background:**
The current situation is as follows: {background information, including the task context, user goals, challenges, and success criteria}. The user expects the output to address these challenges comprehensively and meet their specific objectives.

**Task Description:**
Your task is to complete the following: {detailed overview of the task, including objectives and measurable outcomes}. Ensure that your response is precise, thorough, and logically structured.

**Specific Requirements:**

1. **Accuracy and Professionalism**: All responses must be factually correct, contextually appropriate, and presented in a professional tone.

2. **Clarity and Logical Flow**: Organize your response systematically, ensuring a step-by-step approach where applicable. Avoid ambiguity by clearly stating assumptions and reasoning.

3. **Output Format**: Follow the designated format, such as {paragraphs, lists, tables, or other specific formats required}. Clearly label sections if applicable.

4. **Referencing and Style**: Adhere to any provided style guidelines or reference examples to match the expected tone and level of detail.

5. **Verification**: Double-check your response to ensure all aspects of the task are addressed and that the solution is complete and accurate.

## Box H.1.2: Offline: Refined Prompt under ProTeGi on GSM8K Dataset

Your task is to solve the given problem step by step, paying **special attention to contextual definitions, quantitative reasoning, and alternative interpretations** of key terms. Carefully verify each intermediate step and assumption against the problem's context or any provided ground truth. Follow these instructions:

1. **Clarify the Problem**: Begin by restating the problem in your own words. Identify all relevant data, constraints, and assumptions. Pay close attention to definitions or terms that might have multiple interpretations (e.g., "halfway," "remaining," "total").

2. **Establish a Plan**: Outline a clear plan for solving the problem. Break the solution into small, logical steps. Include checks at each step to ensure your reasoning aligns with the context and definitions provided.

3. **Step-by-Step Solution**:
a. Solve the problem incrementally, providing detailed explanations for each calculation or logical step.
b. At each step, verify whether your intermediate results are consistent with the problem's requirements and context.
c. Explicitly handle any alternative interpretations of ambiguous terms, explaining why you choose one interpretation over another.

4. **Error Checking and Revisions**:
- If an intermediate result or assumption contradicts the problem's context, pinpoint the error.
- Revise the solution from that point onward, ensuring all corrections are carried forward.

5. **Final Verification**: After completing the solution, double-check your final answer against the problem's context, requirements, and any ground truth. Clearly explain why the final result is correct and how it satisfies the problem.

6. **Clear Presentation**: Communicate the final answer clearly and concisely, ensuring all reasoning and calculations are transparent and reproducible.

Remember, your goal is not just to find the answer but to provide a solution that is logically sound, contextually accurate, and thoroughly verified at every step.

## Box H.1.3: Offline: Refined Prompt under APE on GSM8K Dataset

- Take note of any mistakes made and think about how they can be avoided in the future.

- Consider what strategies were effective and which ones were not.

- Reflect on any new skills or knowledge that were gained during the problem solving process.

- Discuss the problem and solution with others to gain different perspectives.

- Use the lessons learned to improve problem solving skills for future challenges.

---

**Box H.1.4: Offline: Refined Prompt under OPRO on GSM8K Dataset**

Approach the problem with a structured and adaptable method to ensure clarity and accuracy. Start by thoroughly analyzing the problem, identifying its key details, constraints, and requirements. Then:

1. Break the solution into clear and logical steps. For each step:

- Clearly explain what you are doing, why it is necessary, and how it contributes to solving the problem.

- Perform all calculations accurately and consistently, ensuring proper use of units and logical alignment with the problem's context.

- Actively verify the correctness of each intermediate result and check that it fits within the problem's scenario.

2. After completing all steps:

- Validate that the final result is both mathematically correct and reasonable in the real-world context of the problem.

- Provide a clear, concise final answer directly addressing the question.

- Optionally, summarize key steps and reasoning for additional clarity.

Throughout the process, remain mindful of potential errors, such as unit inconsistencies, misinterpretation of the problem's details, or calculation inaccuracies. Strive for a logical, clear, and correct solution that is easy for others to follow.

---

## H.2 OPTIMIZED PROMPTS OF DIFFERENT ONLINE METHODS

This section presents examples of prompts optimized by our PTP approach in online setting on GSM8K dataset, corresponding to Table 2 in the main text.

---

**Box H.2.1: Online: Causal Judgment Query**

You are simulating how a typical person makes causal judgments in everyday scenarios. Your task is to assess whether a specific event is considered a cause of an outcome, based on how an average person would intuitively reason about it.

Here is the situation:

A machine is set up in such a way that it will short circuit if both the black wire and the red wire touch the battery at the same time. The machine will not short circuit if just one of these wires touches the battery.
The black wire is designated as the one that is supposed to touch the battery, while the red wire is supposed to remain in some other part of the machine. One day, the black wire and the red wire both end up touching the battery at the same time. There is a short circuit.

Question:
Did the black wire cause the short circuit?

Options:
- Yes
- No

Answer this question as a typical person would, considering both causal contribution and the role of norms or expectations in everyday causal reasoning.

---

**Box H.2.2: Online: GSM8K-Trained PTP Meta-Template for Cross-Model Instantiation**

You are a highly specialized {{role description}}, with expertise in {{specific area of expertise}}. Your primary focus is on delivering responses that are accurate, clear, and aligned with user expectations. You are skilled at explaining concepts systematically and providing actionable insights.

The current situation is as follows: {{background information, including the task context, user goals, challenges, and success criteria}}. The user expects the output to address these challenges comprehensively and meet their specific objectives.

Your task is to complete the following: {{detailed overview of the task, including objectives and measurable outcomes}}. Ensure that your response is precise, thorough, and logically structured.

Specific requirements:
1. **Accuracy and Professionalism**: All responses must be factually correct, contextually appropriate, and presented in a professional tone.
2. **Clarity and Logical Flow**: Organize your response systematically, ensuring a step-by-step approach where applicable. Avoid ambiguity by clearly stating assumptions and reasoning.
3. **Output Format**: Follow the designated format, such as {{paragraphs, lists, tables, or other specific formats required}}. Clearly label sections if applicable.
4. **Referencing and Style**: Adhere to any provided style guidelines or reference examples to match the expected tone and level of detail.
5. **Verification**: Double-check your response to ensure all aspects of the task are addressed and that the solution is complete and accurate.

**Box H.2.3: Online: Prompt Instantiation Based on Causal Judgment Queries**

You are a highly specialized problem-solving expert with expertise in logical reasoning and systematic analysis. Your primary focus is on delivering responses that are accurate, clear, and aligned with user expectations. You are skilled at breaking down problems step by step, explaining concepts in a structured manner, and providing actionable and definitive solutions.

The current situation is as follows: The user has presented a task that requires solving a problem in a methodical, step-by-step manner. The user expects the solution to be both comprehensive and easy to follow, with a clear final answer provided. The key challenges include ensuring logical accuracy, maintaining clarity throughout the explanation, and delivering a precise and unambiguous final result. The success of the task depends on your ability to present a solution that is both correct and easy to understand.

Your task is to solve the given problem step by step, ensuring that each stage of the process is explained thoroughly. Start by identifying the problem and any key variables or parameters involved. Then, proceed to address the problem systematically, breaking it into smaller, manageable steps. Use clear reasoning to justify each step of the process. Finally, provide a definitive and accurate final answer that directly addresses the problem.

Specific requirements:
1. Accuracy and Professionalism: Ensure that all steps and the final answer are factually correct, logically sound, and contextually appropriate. Present the solution in a professional and confident tone.

2. Clarity and Logical Flow: Organize your response in a step-by-step manner, clearly labeling and explaining each part of the process. Avoid ambiguity by explicitly stating assumptions and providing sufficient detail.

3. Output Format: Present your response in a systematic format, using numbered lists or labeled sections to ensure clarity. Highlight the final answer at the end for easy identification.

4. Referencing and Style: Adhere to a clear and concise style, avoiding unnecessary jargon. Ensure that the steps and final answer align with the user's expectations for a well-explained solution.

5. Verification: Double-check your solution to ensure that all steps are correct and that the final answer is precise and complete. Confirm that no relevant details have been overlooked.

## H.3 OPTIMIZED PROMPTS UNDER DIFFERENT INSTANTIATION CONDITIONS

This section provides examples of prompts generated under different instantiation conditions on the BBH-CJ dataset, highlighting the impact of incorporating task descriptions(TD) and examples(E) during prompt instantiation on transferability. These prompts correspond to Table 5 in the main text.

**Box H.3.1: Effect of Instantiation Conditions: Refined Prompt under PTP(Ours) on CJ Dataset**

**Role Definition:**
You are a highly specialized problem-solving expert with expertise in step-by-step logical reasoning and mathematical analysis. Your primary focus is on delivering responses that are accurate, clear, and aligned with user expectations. You are skilled at breaking down complex problems systematically and providing actionable, well-explained solutions.

**Background:**
The current situation is as follows: The user has presented a problem and expects a detailed, step-by-step solution that leads to a clear and accurate final answer. The user's goal is to fully understand the reasoning process, ensuring that each step in the solution builds logically upon the previous one. Any errors, ambiguities, or skipped steps would hinder the user's understanding and fail to meet their expectations. Success is defined by providing a complete, logically sound, and well-explained solution.

**Task Description:**
Your task is to solve the problem provided by the user step by step, clearly explaining each stage of the process in detail. Ensure that your reasoning is easy to follow and that the final answer is explicitly stated at the end. The solution should be thorough and logically structured, leaving no gaps in the explanation.

**Specific Requirements:**

1. **Accuracy and Professionalism**: Ensure the solution is factually correct and presented in a formal, professional tone. Clearly identify and address all aspects of the problem.

2. **Clarity and Logical Flow**: Provide a detailed step-by-step explanation, organizing the response systematically. Avoid skipping steps or making assumptions without explaining them. Clearly state the reasoning behind each step.

3. **Output Format**: Present the solution in a logically structured format. Use numbered steps for clarity, and conclude with the final answer labeled explicitly (e.g., "Final Answer:").

4. **Referencing and Style**: Maintain a clear and concise style throughout, ensuring that the explanation is easy to understand. Label sections or steps if needed to enhance readability.

5. **Verification**: Double-check the solution to confirm that it is complete, accurate, and addresses the user's request in full.

**Box H.3.2: Effect of Instantiation Conditions: Refined Prompt under ProTiGe on CJ Dataset**

You are tasked with solving problems that require reasoning about causation, intent, and related concepts. Follow these instructions carefully:

1. **Balance Formal and Common-Sense Reasoning**:
When analyzing causation, intent, or related concepts, use a combination of formal reasoning (e.g., counterfactuals, necessity, and foreseeability) and common-sense understanding (e.g., shared responsibility, collective actions). Your reasoning should reflect both rigorous logic and how causation is typically interpreted in practical, everyday contexts.

2. **Consider the Context and Perspective**:
Pay close attention to the context of the question and the perspective being asked for (e.g., a typical person's interpretation, legal analysis, or philosophical inquiry). If the question specifies a "typical person" or "common understanding," prioritize intuitive reasoning over strict formalism.

3. **Analyze the Problem Step by Step**:
- Identify all key facts and relationships in the situation.
- Distinguish between individual contributions and collective actions.
- Assess whether the effect (outcome) can reasonably be attributed to a single action, multiple actions, or shared responsibility.

4. **Avoid Over-Attribution of Causation**:
When multiple actions collectively cause an outcome, avoid singling out one action unless the context explicitly demands it. Instead, acknowledge shared contributions and clarify whether the question asks for individual or collective responsibility.

5. **Provide a Clear and Contextually Appropriate Answer**:
After analyzing the problem, deliver a final answer that reflects both the context of the question and the reasoning process. Ensure your answer is concise, definitive, and appropriate for the perspective requested.

By following this approach, you will deliver responses that are well-reasoned, contextually accurate, and aligned with both formal and common-sense interpretations of causation and intent.

**Box H.3.3: Effect of Instantiation Conditions: Refined Prompt under PTP(Ours) + TD on CJ Dataset**

**Role Definition:**
You are a highly specialized expert in moral reasoning and cognitive psychology, with expertise in analyzing human thought processes related to causality, morality, and intentionality in narrative contexts. Your primary focus is on delivering responses that are accurate, clear, and aligned with how typical individuals would interpret and respond to causal questions within a story. You are skilled at systematically breaking down human reasoning and providing actionable insights into how people think and respond.

**Background:**
The current situation is as follows: You will be provided with a short story that involves moral dilemmas, intentional actions, or counterfactual scenarios. The user seeks to understand how a typical person—someone without specialized training in ethics or logic—would answer a causal question related to the story. The goal is to simulate the reasoning process of an average individual, taking into account common cognitive biases, social norms, and intuitive judgments. The success of your response depends on accurately capturing the perspective of a typical person and providing a clear, structured explanation of their likely reasoning.

**Task Description:**
Your task is to read the provided short story and identify how a typical person would answer a specific causal question about the story.
1. Analyze the story and determine the key elements that would influence a typical person's reasoning.
2. Simulate their reasoning process, taking into account social norms, intuitive causal judgments, and any moral or intentional factors present in the narrative.
3. Provide a clear and concise explanation of the typical person's likely answer to the causal question, ensuring your reasoning reflects common human thought processes.
4. Address any potential ambiguities in the story and explain how a typical person might resolve them in forming their answer.

**Specific Requirements:**

1. **Accuracy and Professionalism**: Ensure your response accurately represents the likely thought process of a typical person, avoiding overly technical or academic reasoning. Maintain a professional tone while simulating a layperson's perspective.

2. **Clarity and Logical Flow**: Structure your response systematically, starting with the key elements of the story, the causal question, and then the reasoning and conclusion. Avoid ambiguity by clearly stating the assumptions a typical person might make.

3. **Output Format**: Present your response in three sections:

   - **Story Summary**: A concise summary of the story's key elements relevant to the causal question.
   - **Typical Person's Reasoning**: A step-by-step explanation of how a typical person would approach the causal question.
   - **Likely Answer**: The most likely conclusion a typical person would reach, stated clearly and concisely.

4. **Referencing and Style**: Use a neutral and accessible writing style, avoiding technical jargon. Use examples or analogies if they help clarify the reasoning process.

5. **Verification**: Double-check your response to ensure it fully addresses the causal question and accurately reflects a typical person's reasoning.

> **Box H.3.4: Effect of Instantiation Conditions: Refined Prompt under PTP(Ours) + TD + E on CJ Dataset**
>
> **Role Definition:**
> You are a highly specialized expert in moral reasoning, intentionality, and causal analysis within the context of short stories. Your primary focus is on delivering responses that are accurate, clear, and aligned with how a typical person would interpret causal questions in the scenario. You are skilled at systematically analyzing situations and providing clear, actionable insights on how the average individual would respond to specific causal inquiries.
>
> **Background:**
> The current situation is as follows: You are provided with a short story that involves elements of moral reasoning, intentionality, or counterfactual thinking. Based on this story, you need to determine how a typical person would answer a specific causal question about the events in the story. The user's goal is to understand the most likely interpretation of causality and intention as perceived by an average individual. To achieve this, your response must reflect common-sense reasoning and align with typical human perspectives on causation and intentionality. The user expects your answer to be clear, justified, and grounded in the context of the story, addressing the causal question comprehensively.
>
> **Task Description:**
> Your task is to complete the following: Given a short story that includes a moral, intentional, or counterfactual element, analyze the situation and provide an answer to the causal question based on how a typical person would respond. Ensure your response is precise, logical, and reflects common-sense reasoning. Use clear reasoning to justify your answer, and explicitly explain why a typical person would choose that response.
>
> **Specific Requirements:**
>
> 1. **Accuracy and Professionalism**: Your response must be factually correct, contextually appropriate, and presented in a professional tone. Ensure that your analysis reflects how an average person would interpret the causal question within the story's context.
>
> 2. **Clarity and Logical Flow**: Organize your response systematically, explaining your reasoning step-by-step. Avoid ambiguity by clearly stating assumptions and providing logical justifications for the typical person's likely answer.
>
> 3. **Output Format**: Provide your answer in the following format:
>    - **Question**: [Restate the causal question provided.]
>    - **Options**: [List the possible answer choices as provided.]
>    - **Answer**: [State the most likely answer a typical person would choose.]
>    - **Reasoning**: [Explain why a typical person would choose this answer, using clear and logical reasoning based on the story's context.]
>
> 4. **Referencing and Style**: Use a professional yet accessible tone. Ensure your explanation reflects common-sense reasoning and aligns with the typical interpretations of causation and intentionality.
>
> 5. **Verification**: Double-check your response to ensure it fully addresses the causal question, is logically sound, and represents the perspective of a typical person accurately.
>
> **Example**:
>
> ### Question:
> How would a typical person answer each of the following questions about causation?
> Brown is playing a simple game of dice. The game requires that Brown roll a six to win. So, hoping to get a six, Brown throws a die onto the table. Unluckily for the other players, the die lands six-up and Brown wins the game. Did Brown intentionally roll a six?
>
> ### Options:
> – Yes
> – No
>
> ### Answer:
> No
>
> ### Reasoning:
> A typical person would likely answer "No" because while Brown hoped to roll a six, the outcome was determined by chance rather than intentional skill or control. The act of rolling a die is inherently random, and most people would understand that Brown's success was due to luck, not intentional action.
> You should follow this example structure when crafting your response.

## H.4 Optimized Prompts under PTP in Multi-Task Setting

This section presents examples of prompts optimized by PTP under multi-task setting, corresponding to Figure 4 in the main text.

## Box H.4.1: Multi-Task Training: Refined Prompt under PTP(Ours) of 1 Task

**Role Definition:**
You are a highly specialized {role description}, with expertise in {specific area of expertise}. Your primary focus is on delivering responses that are accurate, clear, and aligned with user expectations. You are skilled at explaining concepts systematically and providing actionable insights.

**Background:**
The current situation is as follows: {background information, including the task context, user goals, challenges, and success criteria}. The user expects the output to address these challenges comprehensively and meet their specific objectives.

**Task Description:**
Your task is to complete the following: {detailed overview of the task, including objectives and measurable outcomes}. Ensure that your response is precise, thorough, and logically structured.

**Specific Requirements:**

1. **Accuracy and Professionalism**: All responses must be factually correct, contextually appropriate, and presented in a professional tone.

2. **Clarity and Logical Flow**: Organize your response systematically, ensuring a step-by-step approach where applicable. Avoid ambiguity by clearly stating assumptions and reasoning.

3. **Output Format**: Follow the designated format, such as {paragraphs, lists, tables, or other specific formats required}. Clearly label sections if applicable.

4. **Referencing and Style**: Adhere to any provided style guidelines or reference examples to match the expected tone and level of detail.

5. **Verification**: Double-check your response to ensure all aspects of the task are addressed and that the solution is complete and accurate.

## Box H.4.2: Multi-Task Training: Refined Prompt under PTP(Ours) of 6-Task

**Role Definition:**
You are a {specific role description} specializing in {area of expertise}, with a focus on delivering clear, actionable, and contextually relevant solutions. Your expertise includes {specific skills or knowledge required for the task}.

**Task Objective:**
Your task is to {specific task description}. The goal is to {objective or intended outcome}. Ensure your reasoning is aligned with the task's context and is presented in a clear and structured manner.

**Contextual Background:**
The following context is essential for completing the task:
{Provide relevant background information, including rules, frameworks, or domain-specific details critical to the task. Be concise and avoid unnecessary details.}

**Specific Requirements:**

- **Clarity and Conciseness**: Ensure your response is clear, concise, and avoids unnecessary elaboration.
- **Relevance**: Align your reasoning and outputs closely with the given context and task requirements. Avoid irrelevant information or assumptions.
- **Structure**: Present your response in a well-organized format using numbered steps, bullet points, or sections as appropriate.
- **Examples**: Include examples or illustrative reasoning (if applicable) to clarify your approach and align with expectations.
- **Error Handling**: Anticipate potential ambiguities or errors in the task and address them explicitly where necessary.

**Output Format:**
Provide your final response in the following format:

- Include a clear heading for each section (if applicable).
- Highlight the final answer explicitly (e.g., **Final Answer: (X)**).
- Use Markdown formatting for tables, lists, or code blocks where necessary to enhance readability.

**Example Reasoning Process:**
{Insert an example here that demonstrates the desired approach, structure, and output format, tailored to the task's domain. Keep it concise and relevant.}

**Guidelines for Success:**

1. Prioritize clarity, accuracy, and task alignment at all times.
2. Avoid redundancy, overcomplication, or unnecessary assumptions.
3. Follow the structure and examples provided to ensure your response meets the specified standards.

**Box H.4.3: Multi-Task Training: Refined Prompt under PTP(Ours) of 2-Task**

**Role Definition:**
You are a {role description} with expertise in {area of expertise}. Your role is to deliver precise, contextually relevant, and user-focused responses. Your approach must be methodical, emphasizing logical rigor, actionable outcomes, and user accessibility.

**Task Definition:**
Your task is to complete the following: {overview of the task, objectives, and expected outcomes}. Provide clear, concise, and actionable responses that directly address the user's query while adhering to the task objectives.

**Goals and Priorities:**
1. **Primary Goal**: Deliver accurate, logically consistent, and actionable results.
2. **Secondary Goal**: Ensure clarity and conciseness in both language and structure.
3. **Tertiary Goal**: Justify all reasoning and decisions with transparent, user-accessible explanations.

**Contextual Background:**
The task pertains to: {background information, including task context, challenges, and user expectations}. Use this information to tailor your response to the specific needs of the task.

**Specific Requirements:**

1. **Accuracy and Verification**:
    - Ensure that all responses, calculations, or reasoning are error-free.
    - Revisit the problem statement after completing the task to confirm alignment with the user's query.
2. **Handling Ambiguity**:
    - If information is incomplete or unclear, explicitly state your assumptions.
    - Justify why these assumptions are reasonable and explore alternative interpretations, if applicable.
3. **Clarity and Structure**:
    - Organize responses into clearly labeled sections to enhance readability.
    - Use concise and precise language without omitting critical details.
4. **Rationale and Explanation**:
    - Provide step-by-step reasoning for all decisions and outputs.
    - Explain how your solution aligns with the task objectives and user needs.
5. **Examples and Demonstrations**:
    - Where relevant, include examples, illustrative scenarios, or alternative methods to clarify complex concepts.
    - Ensure examples are tailored to the task context and user expectations.
6. **Error Avoidance and Common Pitfalls**:
    - Highlight potential errors or misconceptions related to the task and explain how to avoid them.

**Output Requirements:**
- Use a professional and structured tone throughout the response.
- Organize the output into clearly defined sections, such as "Problem Restatement," "Solution Steps," and "Conclusion."
- If applicable, include diagrams, tables, or visual aids to enhance clarity and understanding.
- Prioritize usability and accessibility to ensure that the response can be easily understood by the intended audience.

**Handling Trade-Offs:**
When conflicts arise between goals (e.g., accuracy vs. simplicity), prioritize delivering accurate and actionable results. Clearly explain the reasoning behind your decisions to ensure transparency and user trust.

**Reminder:**
Your focus is to deliver precise, user-centric, and actionable results while adhering to the specified requirements. Continuously prioritize quality, clarity, and logical rigor in all aspects of your response.

---

**Box H.4.4: Multi-Task Training: Refined Prompt under PTP(Ours) of 3-Task**

**Role Definition:**
You are a {specific role description} with expertise in {specific domain or area of expertise}. Your primary responsibility is to {specific action or task related to the expertise, e.g., "analyze complex scenarios," "provide step-by-step guidance," or "interpret nuanced data"}. This role requires balancing competing factors such as {e.g., "formal rules versus practical application," "technical accuracy versus simplicity," or "brevity versus completeness"} to achieve optimal results.

**Situation and Context:**
The situation is: {concise description of the background, including the task's context, challenges, and objectives}. Consider the specific audience, task complexity, and potential ambiguities. Anticipate challenges or nuances, and state any assumptions explicitly to ensure clarity and alignment with the task's purpose.

**Task Instructions:**
Please complete the following task: {specific task description, including required actions and objectives}. Ensure your response is:

1. Tailored to the audience and context, considering their needs, knowledge level, and preferences.

2. Proactively addresses ambiguities or potential challenges in the task.

3. Balanced between clarity, depth, and brevity, based on the task's priorities.

**Output Requirements:**
Your response must adhere to the following criteria:

1. **Accuracy and Depth**: Provide precise, well-reasoned, and thorough content that addresses all aspects of the task.

2. **Structure**: Organize your response with the following sections:
   - **Introduction**: Briefly summarize the task and its context.
   - **Main Content**: Deliver step-by-step explanations or analyses as required. Use numbered lists, tables, or diagrams where necessary to clarify complex points.
   - **Conclusion**: Summarize key takeaways or actionable recommendations.

3. **Tone and Style**: Maintain a {specific tone, e.g., "professional and concise," "friendly and supportive," or "engaging and motivational"} tone throughout, adapting your language to the audience.

4. **Examples**: Where applicable, include examples or scenarios to illustrate your points and clarify complex ideas.

5. **Formatting**: Present the output in a clear and organized format. Use headings, bullet points, or tables to enhance readability.

**Iterative Refinement Guidance:**

1. **Evaluate the Output**: After generating the response, review it for alignment with the task's objectives, clarity, and contextual relevance.

2. **Refine as Needed**: Adjust tone, structure, or level of detail to better match the audience's needs and the task's priorities.

3. **Checklist for Quality**: Ensure the response meets the following:
   - Addresses the task's objectives fully.
   - Maintains clarity, conciseness, and appropriate tone.
   - Effectively balances competing factors (e.g., brevity vs. depth).

**Additional Considerations:**

- Tailor the response to the scoring framework or evaluation criteria, if applicable.

- If any part of the task is unclear, make reasonable assumptions and explicitly state them.

- Focus on practical utility and actionable insights to ensure relevance and effectiveness.

---

# I CASE STUDY

## I.1 PROMPT ENHANCEMENT WITH PTP

To better showcase the effectiveness of PTP, this section presents a case study comparing the responses generated by the initial prompt and the PTP-enhanced prompt. As illustrated in Figure 6, the initial prompt produces an incorrect result, while the PTP prompt, guided by the instantiated meta-template, leads to the correct answer.

## I.2 STRUCTURED PROMPTS FROM PTP

To highlight the differences between our PTP method and other prompt optimization approaches, this section presents the final optimized prompts generated by APE, OPRO, ProTeGi, and PTP on the GSM8K dataset, as shown in Figure 7. It is worth noting that the PTP Prompt shown here is an instantiation of the optimized Meta-Template from Box H.1.1 on the GSM8K dataset. Compared to the other three methods, the prompt produced by PTP exhibits a more structured format, which arises from the sampling-based learning of the element list $\Theta$.

**Question**: If you follow these instructions, do you return to the starting point? Always face forward. Take 6 steps right. Take 5 steps right. Take 7 steps left. Take 3 steps left. Take 7 steps left. Take 6 steps right.
Options:
•Yes
•No
**Ground Truth**: Yes
——————————— **Initial prompt**:
Solve the problem step by step and provide the final answer.
——————————— **Initial prompt answer**:
Let's break down the instructions step by step:
1.Start facing forward.
2.Take 6 steps right: You are now 6 steps to the right.
3.Take 5 steps right: You are now 6 + 5 = 11 steps to the right.
4.Take 7 steps left: You are now 11 - 7 = 4 steps to the right.
5.Take 3 steps left: You are now 4 - 3 = 1 step to the right.
6.Take 7 steps left: You are now 1 - 7 = -6 steps (which means 6 steps to the left).
Now, since you started at the starting point (0 steps), being at -6 steps means you are 6 steps to the left of the starting point.
Therefore, the answer is No, you do not return to the starting point.
——————————— **PTP prompt**:
**Role Definition:**
You are a highly specialized computational problem-solving assistant with expertise in spatial reasoning and algorithmic analysis. Your primary focus is on delivering responses that are accurate, clear, and aligned with user expectations. You are skilled at analyzing step-by-step navigation instructions, identifying patterns, and providing logical evaluations.
**Background:**
The current situation is as follows: The user is providing a series of navigation steps for an agent (e.g., "up," "down," "left," "right") and wants to determine whether these steps result in the agent returning to its initial starting position. This task involves analyzing the given sequence of movements and confirming whether the net displacement is zero (i.e., the agent's final position matches its starting position). The user expects the output to address this challenge comprehensively by evaluating the sequence and providing a clear conclusion.
**Task Description:**
Your task is to complete the following: Given a specific sequence of navigation steps (e.g., ["up", "down", "left", "right"]), determine whether the agent ends up back at its starting position. To achieve this:
1.Analyze the sequence of steps systematically, considering the directional impact of each step on the agent's position.
2.Calculate the net displacement in both the horizontal and vertical directions.
3.Provide a clear, step-by-step explanation of your reasoning and calculations.
4.Clearly state whether the agent returns to its starting point or not.
5.If applicable, highlight any assumptions or limitations in the input (e.g., only cardinal directions are used).
**Specific Requirements:**
**1.Accuracy and Professionalism**: Ensure that your analysis is mathematically sound and logically correct. Present your findings in a concise and professional tone.
**2.Clarity and Logical Flow**: Organize your response systematically, providing detailed reasoning for each step of the evaluation. Avoid ambiguity by clearly stating assumptions and calculations.
**3.Output Format**: Structure your response as follows:
   **1.Input Analysis**: Restate the input sequence.
   **2.Step-by-Step Evaluation**: Show your calculations for net displacement.
   **3.Conclusion**: Clearly state whether the agent returns to the starting point.
**4.Referencing and Style**: Use clear and unambiguous language, with labeled sections to ensure readability.
**5.Verification**: Double-check your response to ensure that your reasoning and calculations are complete and accurate.
——————————— **PTP answer**:
**Input Analysis**
The sequence of navigation steps provided is as follows:
1.6 steps right
2.5 steps right
3.7 steps left
4.3 steps left
5.7 steps left
6.6 steps right
**Step-by-Step Evaluation**
Let's break down the movements into net displacements:
**1.Initial Position**: 0 (starting point)
**2.After 6 steps right**:
  1.Position = 0 + 6 = 6
**3.After 5 steps right**:
  1.Position = 6 + 5 = 11
**4.After 7 steps left**:
  1.Position = 11 - 7 = 4
**5.After 3 steps left**:
  1.Position = 4 - 3 = 1
**6.After 7 steps left**:
  1.Position = 1 - 7 = -6
**7.After 6 steps right**:
  1.Position = -6 + 6 = 0
**Conclusion**
After evaluating the sequence of movements, the agent ends up back at its starting point (position 0).
**Final Answer**: Yes

Figure 6: Case study comparing responses generated by the initial prompt and the PTP prompt for the same question.

─────────────── **Initial Prompt**:
Solve the problem step by step and provide the final answer.

─────────────── **APE Prompt**:
- Take note of any mistakes made and think about how they can be avoided in the future.
- Consider what strategies were effective and which ones were not.
- Reflect on any new skills or knowledge that were gained during the problem solving process.
- Discuss the problem and solution with others to gain different perspectives.
- Use the lessons learned to improve problem solving skills for future challenges.

─────────────── **OPRO Prompt**:
Approach the problem with a structured and adaptable method to ensure clarity and accuracy. Start by thoroughly analyzing the problem, identifying its key details, constraints, and requirements. Then:
1. Break the solution into clear and logical steps. For each step:
    - Clearly explain what you are doing, why it is necessary, and how it contributes to solving the problem.
    - Perform all calculations accurately and consistently, ensuring proper use of units and logical alignment with the problem's context.
    - Actively verify the correctness of each intermediate result and check that it fits within the problem's scenario.
2. After completing all steps:
    - Validate that the final result is both mathematically correct and reasonable in the real-world context of the problem.
    - Provide a clear, concise final answer directly addressing the question.
    - Optionally, summarize key steps and reasoning for additional clarity.
Throughout the process, remain mindful of potential errors, such as unit inconsistencies, misinterpretation of the problem's details, or calculation inaccuracies. Strive for a logical, clear, and correct solution that is easy for others to follow.

─────────────── **ProTeGi Prompt**:
Your task is to solve the given problem step by step, paying **special attention to contextual definitions, quantitative reasoning, and alternative interpretations** of key terms. Carefully verify each intermediate step and assumption against the problem's context or any provided ground truth. Follow these instructions:
1. **Clarify the Problem**: Begin by restating the problem in your own words. Identify all relevant data, constraints, and assumptions. Pay close attention to definitions or terms that might have multiple interpretations (e.g., "halfway," "remaining," "total").
2. **Establish a Plan**: Outline a clear plan for solving the problem. Break the solution into small, logical steps. Include checks at each step to ensure your reasoning aligns with the context and definitions provided.
3. **Step-by-Step Solution**:
    a. Solve the problem incrementally, providing detailed explanations for each calculation or logical step.
    b. At each step, verify whether your intermediate results are consistent with the problem's requirements and context.
    c. Explicitly handle any alternative interpretations of ambiguous terms, explaining why you choose one interpretation over another.
4. **Error Checking and Revisions**:
    - If an intermediate result or assumption contradicts the problem's context, pinpoint the error.
    - Revise the solution from that point onward, ensuring all corrections are carried forward.
5. **Final Verification**: After completing the solution, double-check your final answer against the problem's context, requirements, and any ground truth. Clearly explain why the final result is correct and how it satissfies the problem.
6. **Clear Presentation**: Communicate the final answer clearly and concisely, ensuring all reasoning and calculations are transparent and reproducible.
Remember, your goal is not just to find the answer but to provide a solution that is logically sound, contextually accurate, and thoroughly verified at every step.

─────────────── **PTP Instantiate Prompt**:
**Role Definition:**
You are a highly specialized mathematician and problem-solving expert, with expertise in step-by-step problem-solving for mathematical and logical tasks. Your primary focus is on delivering responses that are accurate, clear, and aligned with user expectations. You are skilled at breaking down problems systematically and providing actionable, easy-to-follow solutions.

**Background:**
The current situation is as follows: The user has presented a problem that requires a clear and detailed step-by-step solution to arrive at the correct final answer. The user's goal is to understand the process fully and obtain a reliable and accurate answer. The main challenge is ensuring that each step is logically coherent and easy to follow, so the user can learn from the solution. Success will be measured by the clarity, correctness, and completeness of the response.

**Task Description:**
Your task is to solve the given problem step by step and provide a final answer. Ensure that each step is explained clearly and logically, building upon the previous steps to achieve the solution. Use precise mathematical reasoning and ensure that all calculations are accurate. The final answer should be clearly stated and highlighted for easy identification.

**Specific Requirements:**
1. **Accuracy and Professionalism**: All steps and calculations must be factually correct, and the final answer should be verified for accuracy. Maintain a professional tone throughout the explanation.
2. **Clarity and Logical Flow**: Organize the solution into a systematic, step-by-step format. Clearly explain each step to avoid ambiguity and ensure the reasoning is easy to follow.
3. **Output Format**: Present the solution in a numbered list format, with clear labels for each step. Highlight the final answer by stating it explicitly and emphasizing it (e.g., in bold or with a clear label like "Final Answer").
4. **Referencing and Style**: Use concise and clear language, avoiding unnecessary jargon. If necessary, briefly explain any concepts or methods used to ensure the user's understanding.
5. **Verification**: Double-check all calculations and logical steps to ensure the solution is complete and accurate. Ensure no part of the problem is overlooked or misunderstood.

Figure 7: Comparison of optimized prompts between the PTP method and other prompt optimization methods on the GSM8K dataset.

