# OpenReview forum: "Prompting to Prompt: Meta-Template Learning for Transferable Prompt Optimization"
_ICLR.cc/2026/Conference — Submitted to ICLR 2026_

### Official Review · Reviewer_xz77 · 2025-10-29

**Soundness:** 3
**Presentation:** 2
**Contribution:** 3
**Rating:** 4
**Confidence:** 4

**Summary:**

This paper proposes Prompting to Prompt (PTP), a novel meta-learning–inspired framework for prompt optimization that introduces a structured meta-template as an intermediate representation. PTP decomposes prompts into reusable, transferable elements and employs a bi-level optimization process: an inner loop refines task-specific prompts using textual gradients, while an outer loop abstracts cross-task structural patterns to update the meta-template. The method is evaluated across six benchmark datasets under both offline and online settings, demonstrating consistent improvements over baselines.

**Strengths:**

1. PTP represents a meaningful conceptual advance by framing prompt optimization as a meta-learning problem. The introduction of a learnable meta-template, enables systematic decomposition and recombination of prompt components, moving beyond string-level or black-box prompt tuning toward structured, interpretable prompt engineering.
2. The dual-loop mechanism is well-motivated and aligns with established meta-learning principles. The inner loop performs fine-grained, error-driven prompt refinement, while the outer loop captures generalizable structural priors. This separation of task-specific adaptation and cross-task generalization is a key strength.
3. PTP enables plug-and-play prompt generation without retraining or auxiliary models, offering a unified solution for both offline and online settings. The ablation studies on loop iterations, instantiation models, and multi-task training further support its robustness and scalability.

**Weaknesses:**

1. While the empirical results are compelling, the paper offers no theoretical analysis of why the meta-template structure facilitates cross-task transfer. There is no discussion of convergence properties, generalization bounds, or conditions under which the bi-level optimization is guaranteed to improve transferability. Given the reliance on LLM-as-optimizer (a non-convex, stochastic process), even heuristic convergence analysis would strengthen the methodological foundation.
2. The 12-element list (Appendix E.1) appears to be manually curated based on prompt engineering heuristics rather than learned from data. The paper does not justify why these specific components (e.g., “Personality & Style,” “Insight & Depth”) are necessary or sufficient. It remains unclear whether $\theta$ is task-agnostic or whether its completeness affects performance—especially for tasks that do not naturally align with such a structured format (e.g., open-ended creative writing).
3. Although PTP is described as “low-cost,” the paper lacks a detailed comparison of computational or financial overhead. All methods are run for 16 total iterations, but PTP uses GPT-4o as M_opt in both loops, while baselines may use cheaper models. A breakdown of API calls, token usage, or wall-clock time would clarify whether the performance gains come at a prohibitive cost—particularly relevant for real-world deployment.
4. The paper demonstrates strong cross-task transfer but does not explore failure modes. For instance:
    - How does PTP perform when source and target tasks are highly dissimilar (e.g., training on arithmetic, testing on moral reasoning)?
    - Can the meta-template resolve conflicting requirements (e.g., “be concise” vs. “provide detailed steps”)?
    - Is PTP effective for tasks that do not benefit from structured prompting (e.g., summarization, translation)?

    Such analyses would better delineate the method’s applicability scope.

5. PTP relies on GPT-4o as M_opt to generate textual gradients. The paper does not evaluate whether weaker models (e.g., GPT-3.5, small open-source LLMs) can serve as effective optimizers. If performance degrades significantly with smaller models, the framework’s practical utility in resource-constrained settings is limited.

**Questions:**

Please refer to Weaknesses for details.

---

> ### Author Response · Authors · 2025-11-19
>
> We thank the reviewer for recognizing the conceptual advance of PTP, particularly its structured meta-template design, dual-loop optimization, and plug-and-play applicability across offline and online settings. We address the concerns raised as follows:
>
> > *W1. While the empirical results are compelling...*
>
> We expand the theoretical discussion in the main method section by framing our approach within meta-learning, often described as "learning to learn." Inspired by this idea, we use bi-level optimization to adapt to individual tasks while refining shared meta-knowledge. The meta-template thus serves as a task-agnostic structural prior that decouples task content from reusable patterns. This meta-learning–inspired design, akin to MAML, is supported by consistent gains across diverse tasks, indicating stable and effective convergence.
>
> Future theoretical directions include information-theoretic generalization bounds for meta-template transfer and stability analysis of structural textual gradient updates to formalize when templates transfer reliably. Full formal proofs remain challenging due to the stochastic, non-convex LLM-as-optimizer setting.
>
> > *W2. The 12-element list (Appendix E.1) appears ...*
>
> We appreciate the reviewer's concern regarding the 12-element list Θ.
>
> 1. The 12 elements were manually curated based on established prompt engineering best practices, covering critical structural components such as role definition and chain-of-thought, which significantly influence LLM output.
> 2. These elements are generally applicable across tasks. For tasks like open-ended creative writing, PTP can still flexibly adapt and extract useful prompts, although the benefits may be smaller compared to structured reasoning tasks.
>
> > *W3. Although PTP is described as “low-cost,”...*
>
> We thank the reviewer for raising this point.
>
> 1. PTP and baselines (e.g., OPRO, ProTeGi) are based on GPT-4o for 16 iterations, so their main training costs in terms of optimizer API calls and iterations are similar.
> 2. The main cost difference is transfer time. PTP's learned meta-template can be applied on a new task with a single instantiation step using the task description and a few optional examples, without collecting new training data and running any optimization loops. In contrast, baselines require task-specific online optimization for every new task, which incurs substantial additional API usage. A preliminary comparison is shown below, and we will include a more detailed cost analysis in the final version.
>
> | Method | Opt Model | Iters | Opt API Calls (Train) | Opt API Calls (Transfer) |
> |--------|-----------|-------|------------------------|----------------------------|
> | PTP    | GPT-4o    | 16    | ~844                    | ~3 (validation-only)                           |
> | ProTeGi    | GPT-4o    | 16    | ~772                    | ~772 (full re-optimization)                       |
>
> > *W4. The paper demonstrates strong cross-task...*
>
> We appreciate the reviewer's comments on potential failure modes. PTP adapts to each task through an explicit instantiation process. As formalized in Eq. (3), the instantiation model maps task requirements to the relevant template elements, producing a task-specific prompt without the redesign of the overall template structure.
>
> 1. For highly dissimilar tasks, the instantiated prompts remain aligned with the task description, though performance may be constrained by the inherent gap between task types.
> 2. When requirements conflict(such as brevity versus detail), we can introduce additional constraints in the task description and further refine the meta-template through incremental optimization.
> 3. For tasks that rely less on structured prompts, such as summarization or translation, PTP can help guide the output according to instructions, but the improvement may be smaller.
>
> > *W5. PTP relies on GPT-4o as $M_{opt}$...*
>
> We thank the reviewer for raising the question regarding optimizer model choice.
>
> 1. In PTP, the optimizer $M_{\text{opt}}$ generates textual gradients that drive prompt updates (Eq. 5-8), so its capability directly impacts gradient quality and optimization effectiveness. Prior studies (APO) show that powerful RLHF-tuned models outperform weaker ones and produce more stable improvements.
> 2. While GPT-4o serves as $M_{\text{opt}}$ in our main experiments for optimal performance, smaller models (e.g., GPT-3.5 or open-source LLMs) can be used in constrained settings, though they produce less accurate textual gradients, resulting in suboptimal meta-templates. In such cases, reusing a meta-template $T$ trained with a stronger optimizer is preferable to retraining with a weaker $M_{\text{opt}}$. Once a high-quality meta-template is obtained, instantiation (Eq. 3) requires minimal model capacity, and Table 3 shows that even weaker task models can instantiate it effectively, demonstrating PTP's robustness in deployment.

---

> > ### Author Response · Authors · 2025-11-23
> >
> > As the Rebuttal deadline approaches, we kindly hope to receive your response . Your feedback is invaluable in helping us improve this work, and we truly appreciate your time and consideration.

---

> > > ### Author Response · Authors · 2025-11-28
> > >
> > > Dear reviewer,
> > >
> > > We hope you are doing well. We are sending this short follow-up in case our earlier responses were missed. We would appreciate it if you could take another look at our clarifications, especially the updates for *W2*.
> > >
> > > For *W2*, we added new analysis to show that PTP is not sensitive to the exact size of the element list Θ. Once Θ reaches a moderate size (about 8–12 elements), the performance becomes stable. This suggests that PTP works because it organizes and activates these elements, not because of how many elements there are. Our preliminary Arena-Hard results are shown below:
> > >
> > > | Metric / Element List Size | 4    | 8    | 12   | 16   |
> > > | -------------------------- | ---- | ---- | ---- | ---- |
> > > | Accuracy                   | 75.5 | 80.7 | 81.8 | 82.2 |
> > >
> > > We appreciate your time and attention, and we would be happy to provide any additional clarifications if needed.
> > >
> > > Best regards,

---

### Official Review · Reviewer_zxZf · 2025-10-31

**Soundness:** 3
**Presentation:** 3
**Contribution:** 3
**Rating:** 4
**Confidence:** 4

**Summary:**

This paper addresses the limited transferability and poor reusability of existing prompt optimization methods for LLMs. The authors propose Prompting to Prompt (PTP), a meta-template learning framework that decomposes prompts into transferable structural elements (meta-templates) and adopts a bi-level optimization process (inner loop for task-specific prompt refinement, outer loop for cross-task meta-template updating). PTP unifies offline and online prompt optimization scenarios, eliminating the need for retraining when adapting to new tasks or queries. Extensive experiments on six benchmark datasets and six LLMs illustrate the state-of-the-art performance with up to 10.52-point gains on challenging tasks like Arena-hard.

**Strengths:**

**Originality**
This work demonstrates originality in two folds: (1) the proposed meta template learning unifies online and offline prompt optimization in one framework. (2) SOTA performances are achieved in extensive experiments.

**Technical details**
1. The reusable meta-templates capture cross-task structural patterns play an important role in improving the transferability.
2. The mathematical equations clearly formalize the prompt optimization objectives and concrete steps in both inner loop and outer loop optimization.

**Quality & Clarity**
1. The proposed method is technically sound and the unified meta learning framework is elegant. The overall structure of the manuscript is organized with a clear logic. Details on the meta learning and bi-level optimization is rigorously addressed in method section with comprehensive supplementary details in Appendix.
2. Extensive experiments with 6 LLMs on 3 open-ended benchmarks, including both offline and online settings, validate PTP's effectiveness. Furthermore, comprehensive datasets and experiment details in appendices provide strong support for this work.

**Significance**
1. This work is of highly practical value to a wide-range of scenarios because of three aspects: (1) the meta template learning is training data efficient (200 instances from GSM-8k or 74 instances from BBH-CJ) , (2) it costs much less GPU resources than previous soft prompt optimization methods, (3) it has the potential to be compatible with nearly all kind of LLMs including both proprietary LLMs with API access and open-sourced LLMs.

**Weaknesses:**

1. Typo in Eq. (4) Select -> Selcet?

2. Previous work PROmpting (OPRO) also introduced concept and method on "meta prompts" to improve the cross-task generalization, as mentioned in Line 160, and the authors claim the superiority of this work with only a few words in between Line 186-187. Since OPRO is the most related work, quantitative and qualitative comparisons between the learned meta-templates and meta prompts in OPRO should be carefully performed and discussed.

3. I’m curious about the origin of the term 'text gradient'—where does it come from? Since the text gradients are not real gradients, how could the authors assure their claim in Line 188 "This data-driven extension retains the generalization ... while further improving adaptability in ..."

4. The element list Θ (Appendix E.1) includes 12 components, but the paper does not explain how these components were selected? How would the total number of the list influence the overall results?

5. In Table 2, PTP’s average gain over P3 is 10.52 points on Arena-hard but only 0.12-0.25 points on Alpaca-Eval-2.0 for GPT-4-turbo and GPT-4-1106-preview. The paper attributes this to "task differences" but provides no further analysis. What is the main reason that the difference in tasks/model capabilities cause such huge performance gaps?

**Questions:**

Several questions are raised in "Weaknesses" part. The main concern is the lack of detailed analysis and comparisons with most related work OPRO. I would be willing to change my recommendation according to the authors' response.

---

> ### Author Response · Authors · 2025-11-19
>
> We thank the reviewer for highlighting the originality of our meta-template framework, its technical rigor, and the practical value demonstrated through extensive experiments across multiple LLMs and benchmarks. We address the concerns raised as follows:
>
> > *W1. Typo in Eq. (4) Select -> Selcet?*
>
> We thank the reviewer for pointing out the typo in Eq. (4). The word “Selcet” in Line 236 has been corrected to “Select” in the revised manuscript and we have proofread the revised manuscript carefully.
>
> > *W2. Previous work PROmpting (OPRO) also introduced concept and method on "meta prompts" to improve...*
>
> We appreciate the reviewer's suggestion and have expanded the comparison with OPRO in the revision.
>
> 1. PTP's meta-template is different from OPRO's meta-prompt basically in purpose and behavior.
>
> * ​OPRO meta-prompt is static: without an optimization target. The meta-prompt is a fixed auxiliary instruction containing past scores/solutions and is used only to help the model propose prompt candidates during optimization. After training, it outputs ​a task-specific prompt​.
> * ​PTP meta-template is optimized​: with the object under iteration​, gradually learning task-agnostic structural priors across tasks. After training, PTP uses the learned template to instantiate task-specific prompts based on the task description/query.
>
> 2. They own different generalization properties.
>
> * OPRO generalizes mainly through the fixed auxiliary prompt and tends to overfit to optimization traces of the training tasks.
> * PTP explicitly separates structure (learned in the meta-template) from content (injected only at instantiation), with stronger cross-task transfer, validated via our experiments.
>   ​3. The revised manuscript now includes both conceptual and empirical comparisons, clarifying why PTP offers better cross-task robustness and reusability than OPRO.
>
> > *W3. I'm curious about the origin of the term 'text gradient'...*
>
> We have updated the main text to consistently use the term "textual gradient" and added a description of its origin and validity in Section 3.2. We thank the reviewer for asking about it.
>
> 1. The term "textual gradient" is inspired by prior works in automatic prompt optimization and refers to natural-language feedback from LLMs ($M_{opt}$) indicating the directions of improving prompts[1]. It is not a traditional backpropagation gradient.
> 2. Effectiveness is ensured via two mechanisms: in the inner loop, candidate prompts generated according to $g_B$ are evaluated on a validation subset ($\mathcal{D}_{i,\text{val}}$) using Beam Search, and only the updates of improving actual performance are accepted; the outer loop generates structure-oriented gradients $g_S$ to update $T$, enhancing separation between task content and structural priors, which preserves generalization and adaptability.
>
>
>
> > *W4. The element list Θ (Appendix E.1) includes 12 components...*
>
> We appreciate the reviewer's questions regarding the element list $\Theta$.
>
> 1. The 12 elements were manually curated based on established prompt engineering best practices, covering critical structural components such as role definition and chain-of-thought, which significantly influence LLM output.
> 2. $\Theta$ acts as an atomic library for the bi-level optimization. PTP learns how to arrange and activate these elements, rather than merely relying on their presence. Performance is robust as long as $\Theta$ contains the key structural elements.
> 3. Lack of elements limits PTP to explore structural patterns and degrades performance. However, adding redundant elements can only provide minimal improvement. Hence, PTP's core value lies in learning structure and contextualize elements rather than simply increasing their number. Our preliminary results on Arena-hard are shown below:
> | Metric / Element List Size    | 4 | 8 | 12 | 16 |
> | ------------------ | ------------ | ------------ | ------------- | ------------- |
> | Accuracy | 75.5       | 80.7       | 81.8        | 82.2        |
>
>
>
> > *W5. In Table 2, PTP's average gain over P3 is 10.52 points on Arena-hard but...*
>
> We thank the reviewer for highlighting performance differences among tasks.
>
> 1. Arena-hard involves complex multi-step reasoning and structured instructions. PTP's meta-template $T^*$ captures structural priors (e.g., enforcing CoT), leading to a significant gain of 10.52 points over P3.
> 2. Alpaca-Eval-2.0 consists of simpler instruction-following tasks where GPT-4-turbo and GPT-4-1106-preview already perform near the state-of-the-art. Structural optimization via PTP contributes minimally, resulting in gains of only 0.12–0.25 points.
> 3. These results indicate that PTP provides the most benefit in tasks with higher complexity, while simpler tasks show limited performance improvement due to the already strong baseline capabilities.
>
> [1]Pryzant, Reid, et al. "Automatic prompt optimization with "gradient descent" and beam search." arXiv preprint arXiv:2305.03495 (2023).

---

> > ### Author Response · Authors · 2025-11-23
> >
> > As the Rebuttal deadline approaches, we kindly hope to receive your response . Your feedback is invaluable in helping us improve this work, and we truly appreciate your time and consideration.

---

> ### Comment · Reviewer_zxZf · 2025-11-26
>
> Thanks for the authors' response. Most of my concerns are clearly addressed, and my rating has been changed to "weak accept".

---

### Official Review · Reviewer_Lbq7 · 2025-10-31

**Soundness:** 2
**Presentation:** 3
**Contribution:** 2
**Rating:** 2
**Confidence:** 4

**Summary:**

This paper introduces a new Prompting to Prompt (PTP) framework for prompt optimization following the ideas of meta learning. It focuses on improving the disadvantages of the existing prompting methods, that they usually have poor transferability across tasks and a strong dependency on task-specific data. In PTP, a highly structured meta template/prompt is optimized through a bi-level optimization process. The inner loop adapts the meta template into task-specific prompts utilizing the sample-level errors, then the outer loop extracts important structural changes that are general enough across tasks. The paper shows empirical improvements in performance in both offline and online prompt optimization scenarios.

**Strengths:**

1. The meta prompting formulation enables application in both offline and online settings - the meta prompt template can serve as task-level prompts, and can also adapt to query-level prompts.
2. Transferability is the key to this paper. The paper shows strong generalization abilities to different tasks and models empirically.

**Weaknesses:**

1. The method shows dependency to the manually-curated element list for the meta template. More analysis and discussion on relaxing this requirement are needed.
2. The cost of meta training is not discussed in the paper, making the evaluation of the proposed method incomplete.
3. There is a high reliance on the powerful frontier model for optimizing the prompts through textual prompt updates. In the paper, GPT-4o is heavily used.
4. Some parts of the formulation need to be clarified.

I included more details in the questions section below.

**Questions:**

1. In the meta prompting formulation of the paper, should we find the argmax over the tasks $i$ as well? Currently, it is unclear whether the dataset $D = \\{ (D_i, C_i) \\}$ includes multiple tasks and whether they are optimized jointly according to the formulation.
2. Why don’t we need a validation set in the PTP algorithm? I see two potential problems: Firstly, the meta prompt can be easily overfitted to the training dataset, for example, by incorporating task-query-specific information explicitly in the prompt, etc. Secondly, prompt updates (or gradients) for both behavioral and structural prompts are executed by the optimizer LLM, there is no guarantee that the updated prompt outperforms the previous prompt on unless you evaluate it against the validation set.
3. Is the element list for the meta template static? How did you come up with these 12 elements and prompts? This begs two questions: Firstly, whether by explicitly designing the meta prompt using these 12 elements manually by a human achieves similar performance as compared to PTP? Secondly, is the PTP framework naturally extendable by making the LLMs curate the elements list by themselves?
4. The meta-training process seems to be computationally expensive to me. The paper still lacks a comprehensive analysis of the computational cost (and/or the API call budget requirements) of the proposed PTP framework. Preferably, an explicit cost comparison alongside the performance comparison (in Table 1 and Table 2) should be presented.
5. Serious typo in Line 236: “Selcet” → “Select”

---

> ### Author Response · Authors · 2025-11-19
>
> We thank the reviewer for noting that our meta-prompting formulation applies to both offline and online settings and demonstrates transferability across tasks and models. We address the concerns below.
>
> > *Q1. In the meta prompting formulation of the paper, should we find the argmax over the tasks  as well?*
>
> We thank the reviewer for pointing out the potential ambiguity in the meta-prompt formulation regarding multiple tasks.
>
> 1. During training, the dataset $\mathcal{D}$ contains multiple tasks. The expectation over $(x_{ij}, y_{ij}) \sim \mathcal{D}_i$ is intended to jointly optimize $T$ among all tasks. It is unnecessary to add $\arg\max$ over different tasks, because the optimizer will favor a single task and deviate from cross-task generalization.
> 2. PTP is also compatible with single-task datasets. In that case, the task distribution reduces to a single $\mathcal{D}_1$ and the best $T^*$ in the optimization objective will be achieved. The flexibility of PTP is demonstrated while its structural prior learning principle is preserved.
>
> > *Q2. Why don't we need a validation set in the PTP algorithm?*
>
> We appreciate the reviewer's concern regarding validation and potential overfitting.
>
> 1. Validation Set Usage: For each task, the training set is split equally into training and validation subsets. The inner loop generates candidate prompts on the training subset (Eq. 5, 6) and selects top-performing ones via beam search on the validation subset (Algorithm 2). Final results use separate test sets. We have now clarified this data splitting procedure in Appendix C.
> 2. Quality Assurance of LLM-Generated Updates:  Our validation-based beam search addresses gradient quality through: (i) Validation-guided selection – Algorithm 2 filters candidates by validation performance, removing poor-quality updates from LLM errors; (ii) Structural regularization – The meta-template T encodes task-agnostic priors, and outer-loop aggregation across tasks mitigates overfitting to single-task data; (iii) Empirical evidence – Figure 5 shows performance plateaus (not degrades) after K_Inner=4, confirming validation prevents harmful update accumulation.
>
> > *Q3. Is the element list for the meta template static? How did you come up with these 12 elements...*
>
> We thank the reviewer for the thoughtful questions about the element list $\Theta$.
>
> 1. $\Theta$ is a static but extensible set of 12 atomic prompt components, derived from common prompt-engineering practices. This serves as a stable structural space for optimization, much like how neural networks rely on a fixed initialization choice before training[1-2].
> 2. Human experts could manually assemble prompts from these elements, but doing so is labor-intensive, depends heavily on expertise, and typically causes less stable performance. However, PTP sorts elements automatically among tasks and captures structural patterns difficult to tune manually.
> 3. The framework is naturally extensible. Future versions could allow the optimizer LLM to propose or refine elements, enabling more automated management of $\Theta$.
>
>
> > *Q4. The meta-training process seems to be computationally expensive to me...*
>
> We appreciate the reviewer highlighting computational cost considerations.
>
> 1. PTP and baselines (e.g., OPRO, ProTeGi) are based on GPT-4o for 16 iterations, so their main training costs in terms of optimizer API calls and iterations are similar.
> 2. The main cost difference is transfer time. PTP's learned meta-template can be applied on a new task with a single instantiation step using the task description and a few optional examples, without collecting new training data and running any optimization loops. In contrast, baselines require task-specific online optimization for every new task, which incurs substantial additional API usage. A preliminary comparison is shown below:
>
> | Method | Opt Model | Iters | Opt API Calls (Train) | Opt API Calls (Transfer) |
> |--------|-----------|-------|------------------------|----------------------------|
> | PTP    | GPT-4o    | 16    | ~844                    | ~3 (validation-only)                           |
> | ProTeGi    | GPT-4o    | 16    | ~772                    | ~772 (full re-optimization)                       |
>
>
>
> > *Q5. Serious typo in Line 236: “Selcet” → “Select”*
>
> We thank the reviewer for noticing the typo. The word “Selcet” in Line 236 has been corrected to “Select” in the revised manuscript.
>
> [1]Zhang, Yifan, Yang Yuan, and Andrew Chi-Chih Yao. "Meta prompting for ai systems." arXiv preprint arXiv:2311.11482 (2023).
> [2] Wang, Ming, et al. "LangGPT." GitHub Repository, LangGPTai/LangGPT, https://github.com/langgptai/LangGPT (2024).

---

> > ### Author Response · Authors · 2025-11-23
> >
> > As the Rebuttal deadline approaches, we kindly hope to receive your response . Your feedback is invaluable in helping us improve this work, and we truly appreciate your time and consideration.

---

### Official Review · Reviewer_dcxS · 2025-10-31

**Soundness:** 2
**Presentation:** 3
**Contribution:** 3
**Rating:** 6
**Confidence:** 3

**Summary:**

This paper addresses the challenge of transferable prompt optimization for large language models (LLMs), a problem where existing methods tend to overfit to specific tasks and lack generalizability. The authors propose "Prompting to Prompt" (PTP), a bi-level meta-learning framework that optimizes meta-templates—structured, decomposable representations of prompts—enabling systematic reuse and adaptation across offline and online settings. The method incorporates inner-loop prompt refinement based on feedback and an outer-loop update that distills structural knowledge across tasks. Results on six datasets, including arithmetic, reasoning, and alignment challenges, demonstrate strong average gains over a suite of baselines, and ablations provide further insight into transfer mechanisms and design choices.

**Strengths:**

1.The paper comprehensively articulates the need for prompt optimization methods that transfer across tasks, highlighting both empirical and practical shortcomings of current approaches (see Section 1 and Table 1).
2.The effect of meta-template scale, instantiation models, and optimization loop parameters is dissected in Figure 4 and Figure 5. For example, Figure 5 systematically shows how accuracy saturates with increasing inner and outer loop iterations, supporting design choices.

**Weaknesses:**

1. Notation and references are ambiguous—especially defining “prompt elements,” mapping them to meta-templates, and explaining inner vs. outer loop mechanics; clearer legends/footnotes would help.
2. Needs deeper before/after prompt comparisons to show which elements transfer or recur; consider a main-text figure contrasting PTP with ProTeGi/OPRO.
3. Heavy use of API-only LLMs limits technical depth vs. prompt/soft-prompt tuning; misses post-hoc analysis of how meta-templates align with representations or how elements evolve across tasks.
4. Update/search are under-specified (feedback granularity, candidate scoring/beam policy, data flow/complexity) and lack pseudocode; no convergence guarantees, failure-mode diagnostics, or checks for overfitting/negative transfer.

**Questions:**

See Weaknesses.

---

> ### Author Response · Authors · 2025-11-19
>
> We appreciate the reviewer's recognition of our motivation for developing transferable prompt-optimization methods, as well as the reviewer's acknowledgement of our analyses on the effects of meta-template scale and optimization-loop design. We address the reviewer's specific concerns below.
>
> > *W1. Notation and references are ambiguous—especially defining “prompt elements,”...*
>
> We thank the reviewer for highlighting the ambiguity in notation and loop mechanics.
>
> 1. The element set $\Theta$ is a static yet extensible library of atomic prompt components, manually curated from established prompt-engineering practices to capture core structural factors known to influence LLM behavior.  $T$ is a learned structure whose element selection and element descriptions are iteratively optimized during training. $\Theta$ only defines the structural search space; task-specific information is injected later during instantiation.
> 2. The inner loop adapts each task by instantiating $T$, monitoring errors, and using behavioral textual gradients to refine the prompt structure, thereby enhancing performance and guiding structural improvements.
> 3. The outer loop abstracts cross-task structure. By comparing prompts before and after inner-loop adaptation across tasks, it identifies recurring structural improvements and updates $T$ accordingly. Hence, transferable structural priors in $T$ are accumulated to support cross-task generalization.
> 4. The framework figure and its title have been updated to make the roles of $Θ$, $T$, and the two optimization loops easier to follow.
>
> > *W2. Needs deeper before/after prompt comparisons to...*
>
> We appreciate the reviewer's request for a clearer comparison of how prompt elements transfer or recur across tasks.
>
> 1. The appendix already includes examples of the meta-template, instantiated prompts, and outputs from OPRO and ProTeGi. To make the comparison clearer, we will add an additional figure in the appendix H.2 that visualizes the meta-template before and after training and presents it side-by-side with the structures from the baseline methods.
> 2. PTP learns reusable structural patterns rather than task-specific text, which explains its stronger transfer performance compared with methods such as OPRO and ProTeGi that optimize a separate prompt for each task.
>
> > *W3. Heavy use of API-only LLMs limits technical depth...*
>
> We thank the reviewer for raising concerns about working with API-only LLMs and the limitations this may impose.
>
> 1. Many widely used frontier models are only accessible through API interfaces. Designing a method that operates in this setting is therefore aligned with common real-world constraints. Prompt optimization under API restrictions is a practical and relevant direction.
> 2. Even without access to model parameters, textual gradients provide meaningful signals for analysis. PTP uses both content-level and structure-level textual gradients during training, which makes it possible to examine how elements evolve across tasks and how the meta-template aligns with task variations. This offers a degree of interpretability despite the API-only setting.
>
> > *W4. Update/search are under-specified (feedback granularity, candidate scoring...*
>
> We thank the reviewer for highlighting the under-specification of update/search, feedback granularity, and convergence diagnostics.
>
> 1. Feedback is based on batch-level bad-case sampling. These cases guide candidate prompt generation, which is scored on a validation set, and beam search selects the top candidates. We include the detailed evaluation and selection procedure in the Appendix B.
> 2. Convergence is evaluated empirically on validation sets and stable trends among tasks are shown in results (Figure 5) via the bi-level structure.
> 3. The bi-level design mitigates overfitting and negative transfer: the inner loop adapts to task-specific needs, while the outer loop extracts stable cross-task structural patterns.
> 4. For diverse tasks, instantiated prompts follow the task description but performance may be limited. Conflicting requirements, like brevity versus detail, are guided by the task description and adjusted via optimization, though outcomes may not always be optimal.

---

> > ### Author Response · Authors · 2025-11-23
> >
> > As the Rebuttal deadline approaches, we kindly hope to receive your response . Your feedback is invaluable in helping us improve this work, and we truly appreciate your time and consideration.

---

> > ### Comment · Reviewer_dcxS · 2025-11-24
> >
> > Thanks for your response. As the score is already very high, I have decided to maintain it as is.

---

### Meta-Review · Area_Chair_8aTD · 2026-01-03

**Summary:**

The paper proposes "Prompting to Prompt" (PTP), a meta-learning framework for prompt optimization. While reviewers acknowledged the strong empirical performance on benchmarks like Arena-hard and the originality of the unified offline/online framework, significant concerns remain regarding the practicality and theoretical grounding of the method. Specifically, Reviewers Lbq7 and xz77 expressed strong reservations about the framework's heavy reliance on expensive, frontier models (e.g., GPT-4o) for the optimization process and the lack of a comprehensive cost analysis comparing PTP against all relevant baselines. Additionally, the lack of theoretical analysis and failure mode exploration remains a hurdle for a higher assessment.

**Reviewer Concerns:**

**Addressed Concerns:**
* **Comparison with OPRO (Methodological):** The authors clarified the distinction between PTP's optimized meta-template and OPRO's static meta-prompt, which was accepted by Reviewer zxZf.
* **Validation Set Usage:** The authors clarified for Reviewer Lbq7 that a validation set is used during the inner loop beam search, correcting the reviewer's misunderstanding that no validation was performed.
* **Textual Gradients:** The origin and definition of "textual gradients" were explained to the satisfaction of Reviewer zxZf.
* **Element List Size:** Ablation studies on the size of the element list were provided to address concerns from Reviewers zxZf and xz77.

**Outstanding Concerns:**
* **Reliance on Frontier Models:** Reviewers Lbq7 and xz77 raised concerns about the necessity of using powerful models like GPT-4o for the optimizer. The authors admitted that weaker models produce "suboptimal meta-templates," confirming the reliance, which remains a validity concern for resource-constrained settings.
* **API Cost Comparison (specifically OPRO):** Reviewer Lbq7 requested a cost comparison. While the authors provided a table for ProTeGi, they did not explicitly list the cost data for OPRO, which appears to be the most competitive baseline in Table 1.
* **Theoretical Analysis:** Reviewer xz77's request for convergence properties or generalization bounds was not addressed with formal analysis.
* **Failure Modes:** Reviewer xz77's request for analysis of failure modes (e.g., dissimilar tasks, conflicting requirements) was addressed only with high-level discussion rather than empirical evidence.

**Reviewer Scores:**

* **Reviewer dcxS: 6.** The reviewer explicitly stated they decided to maintain their score of 6.
* **Reviewer Lbq7: 4.** Originally a 2. The reviewer's misunderstanding regarding the lack of a validation set was corrected, which warrants an increase. However, the reviewer's major concerns regarding the high reliance on powerful frontier models and the incomplete cost analysis (missing OPRO comparison) remain, preventing a higher score.
* **Reviewer zxZf: 6.** The reviewer explicitly upgraded their score to "weak accept" (6) following the rebuttal.
* **Reviewer xz77: 4.** Originally a 4. This reviewer is unlikely to increase their score because their primary concerns regarding theoretical analysis and failure modes were not empirically addressed. Furthermore, the concerns shared with Reviewer Lbq7 regarding the reliance on GPT-4o and cost overhead remain outstanding.

---

### Decision · Program_Chairs · 2026-01-26

Reject